# MXene-Based Nanocomposites for Piezoelectric and Triboelectric Energy Harvesting Applications

**DOI:** 10.3390/mi14061273

**Published:** 2023-06-20

**Authors:** Durga Prasad Pabba, Mani Satthiyaraju, Ananthakumar Ramasdoss, Pandurengan Sakthivel, Natarajan Chidhambaram, Shanmugasundar Dhanabalan, Carolina Venegas Abarzúa, Mauricio J. Morel, Rednam Udayabhaskar, Ramalinga Viswanathan Mangalaraja, Radhamanohar Aepuru, Sathish-Kumar Kamaraj, Praveen Kumar Murugesan, Arun Thirumurugan

**Affiliations:** 1Departamento de Mecánica, Facultad de Ingeniería, Universidad Tecnologica Metropolitana, Santiago 8330378, Chile; durgaprasad.pabba@gmail.com (D.P.P.); uday.rednam@utem.cl (R.U.); raepuru@utem.cl (R.A.); 2Department of Mechanical Engineering, Kathir College of Engineering, Coimbatore 641062, India; nittsathya@gmail.com; 3School for Advanced Research in Polymers (SARP), Central Institute of Petrochemicals Engineering & Technology (CIPET), T.V.K. Industrial Estate, Guindy, Chennai 600032, India; ananthsharp@gmail.com; 4Centre for Materials Science, Department of Physics, Faculty of Engineering, Karpagam Academy of Higher Education, Coimbatore 641021, India; sakthi1807@gmail.com; 5Department of Physics, Rajah Serfoji Government College (Autonomous), Thanjavur 613005, India; nchidambaraselvan@gmail.com; 6Functional Materials and Microsystems Research Group, RMIT University, Melbourne, VIC 3000, Australia; shanmuga.sundar.dhanabalan@rmit.edu.au; 7Sede Vallenar, Universidad de Atacama, Vallenar 1612178, Chile; carolina.venegas@uda.cl; 8Departamento de Química y Biología, Facultad de Ciencias Naturales, Universidad de Atacama, Copiapó 1531772, Chile; mauricio.morel@uda.cl; 9Faculty of Engineering and Sciences, Universidad Adolfo Ibanez, Santiago 7941169, Chile; mangal@uai.cl (R.V.M.); praveenkumarmurugesan280589@gmail.com (P.K.M.); 10Instituto Politécnico Nacional, Centro de Investigación en Ciencia Aplicada y Tecnología Avanzada, Unidad Altamira (CICATA Altamira), Altamira 89600, Mexico; sathish.k@llano.tecnm.mx

**Keywords:** MXene, energy harvesting, composite materials, piezoelectric, triboelectric nanogenerator

## Abstract

Due to its superior advantages in terms of electronegativity, metallic conductivity, mechanical flexibility, customizable surface chemistry, etc., 2D MXenes for nanogenerators have demonstrated significant progress. In order to push scientific design strategies for the practical application of nanogenerators from the viewpoints of the basic aspect and recent advancements, this systematic review covers the most recent developments of MXenes for nanogenerators in its first section. In the second section, the importance of renewable energy and an introduction to nanogenerators, major classifications, and their working principles are discussed. At the end of this section, various materials used for energy harvesting and frequent combos of MXene with other active materials are described in detail together with the essential framework of nanogenerators. In the third, fourth, and fifth sections, the materials used for nanogenerators, MXene synthesis along with its properties, and MXene nanocomposites with polymeric materials are discussed in detail with the recent progress and challenges for their use in nanogenerator applications. In the sixth section, a thorough discussion of the design strategies and internal improvement mechanisms of MXenes and the composite materials for nanogenerators with 3D printing technologies are presented. Finally, we summarize the key points discussed throughout this review and discuss some thoughts on potential approaches for nanocomposite materials based on MXenes that could be used in nanogenerators for better performance.

## 1. Introduction

The sustainable development of people’s lifestyle needs requires emerging green and renewable energy systems, which are the definite solution for global warming and demanding energy requirements [1]. Moreover, people approach the energy crisis with various personal and social needs, which leads to focus on different kinds of energy harvesting systems [2]. In the environment, mechanical energy is enormously available in the form of minor and major disturbances. This can be easily converted into useful energy to fulfill current energy requirements. Instead of using these energies, for the most part they are fully wasted or only partially used for energy conversion [3,4]. 

Nanogenerator types such as piezoelectric and triboelectric are developed to harvest energy from mechanical movements [5,6]. Here, easily accessible and fabricated materials of ceramics, polymers, their composites, and nanocomposites are used for various locations, ambiences, and applications [7]. Still now, the acquired performance of nanogenerators is comparatively low, which needs to be improved to obtain the highest energy conversion [8]. There is a big gap between practically gained performances due to the inadequate acquired performance of piezoelectric nanogenerators (PENGs). In addition to piezoelectric nanogenerators, there are also other types of nanogenerators such as triboelectric nanogenerators (TENGs), which generate electrical energy from friction and pyroelectric nanogenerators, which generate electrical energy from temperature changes [9,10,11]. Nanogenerators have a number of potential applications in areas such as energy harvesting, self-powered sensors, and wearable electronics [12]. They can be used to generate electrical energy from ambient vibrations, such as those caused by footsteps or wind, or from human movement such as walking or typing [13,14].

Recent studies have shown that MXenes have high mechanical strength, electrical conductivity, and thermal stability, making them ideal for energy harvesting applications [15,16]. In particular, MXene-based composites have been shown to exhibit improved energy conversion efficiency compared to traditional energy harvesting materials [17]. For example, MXene-based composites have been shown to be effective for harvesting mechanical energy from ambient vibrations and for generating electricity from temperature gradients [18,19]. In addition, MXenes have been explored for various energy harvesting methods, such as piezoelectric energy harvesting, electromagnetic wave harvesting, and energy storage applications [20,21,22,23]. Researchers have found that incorporating MXenes into piezoelectric materials can improve the energy conversion efficiency and increase the power output [24]. The results of previous studies have demonstrated the potential of MXene-based energy harvesting for a variety of energy sources [25]. Further research and development in this field is expected to lead to even more efficient and effective energy harvesting technologies in the future [26,27].

The voltage output of MXene-based nanogenerators depends on several factors, including the specific MXene material used, the configuration of the device, and the mechanical energy input [28]. In general, MXene-based nanogenerators have demonstrated high voltage output, with some studies reporting voltage outputs of up to several volts under optimal conditions [29]. The high voltage output of MXene piezoelectric nanogenerators is due to the strong piezoelectric properties of MXene materials, which allow for efficient conversion of mechanical energy into electrical energy [24,30]. Additionally, the voltage output of MXene piezoelectric nanogenerators can be improved by optimizing the device configuration and by using MXene materials with higher piezoelectric coefficients [31,32]. It should be noted that the voltage output of MXene piezoelectric nanogenerators is typically small and is not enough to power most electronic devices on its own [33]. However, the small voltage output can be used to power low-power devices, such as sensors, or can be stored in a battery for later use [34]. MXene piezoelectric nanogenerators have demonstrated promising voltage output and have the potential to be used for applicable energy harvesting applications. There is ongoing research to further optimize the performance and stability of these devices for practical applications [18,35,36].

This paper presents the latest developments in MXene-based materials, especially for triboelectric and piezoelectric nanogenerators. Figure 1 shows the overview of MXene-based materials for energy harvesting. To begin, this review covers the fundamental background of piezoelectric and triboelectric nanogenerators, encompassing their key parameters, working mechanisms, and challenges. Furthermore, this review highlights the superior properties of MXene-based materials for piezoelectric and triboelectric nanogenerators. Subsequently, this review comprehensively discusses the design tactics and mechanisms of MXenes in enhancing the output performance of nanogenerators. Finally, this review offers some perspectives and suggestions on future design tactics for MXene-based composite materials in the development of advanced nanogenerators. 

## 2. Energy Harvesting

The current requirements of renewable energy are focused on several factors: the increasing demand for energy, the need to diminish greenhouse gas emissions, and the desire for energy independence and security [37]. The world’s energy demand is increasing as the global population grows and economies develop. Renewable energy sources, such as solar, wind, and hydropower, are seen as a way to meet this growing demand without relying on finite fossil fuels. Climate change is one of the biggest challenges facing the world today. Hence, renewable energy sources do not produce greenhouse gases, making them an important part of the solution to decreasing emissions and mitigating climate change [38]. Renewable energy sources are often locally sourced, which reduces dependence on foreign energy sources and enhances energy security [39]. The cost of renewable energy technologies has reduced significantly in recent years, making them increasingly competitive with traditional fossil fuels [40]. This has led to increased adoption of renewable energy in both developed and developing countries [41]. Moreover, renewable energy sources can be distributed and decentralized for meeting greater energy access in rural areas and reducing the need for long-distance energy transmission [42]. Anyhow, the current needs of renewable energy are diverse and complex, and meeting these needs will require a combination of technical advances, policy support, and market incentives [43]. However, with continued investment and innovation, renewable energy has the potential to play a major role in meeting the world’s energy needs in a sustainable and responsible manner [44].

### 2.1. Nanogenerator

A nanogenerator is a device that converts mechanical energy into electrical energy on a nanoscale [45]. Nanogenerators operate on the basis of two effects: piezoelectric and triboelectric [46]. A nanogenerator can generate electrical energy by applying mechanical stress to a piezoelectric material [47], and a triboelectric material can generate electrical energy by contact and separation. Moreover, this can be integrated into wearable devices such as smartwatches to generate electrical energy from the wearer’s movement [48]. Then, this energy can be used to power the device to eliminate the need for a battery. Hence, nanogenerators have great potential for developing self-powered devices and for harvesting energy from the environment, making them an exciting area of research and development in the field of energy harvesting [49,50,51].

### 2.2. Working Principle of Piezoelectric Nanogenerators 

The function of a PENG is based on the piezoelectric effect, which is a phenomenon in which electrical charges are generated in certain materials in response to mechanical stress or strain [52,53]. In a piezoelectric nanogenerator, a piezoelectric material (such as lead zirconate titanate (PZT) or zinc oxide (ZnO)) is used as the active layer, which is sandwiched between two conductive electrodes [54,55]. When an external mechanical force is applied to the nanogenerator such as pressure, bending, or vibration, it causes the piezoelectric material to deform and generate an electrical charge. This electrical charge is collected by the electrodes and can be used as a source of energy [56,57,58]. The schematic of the working mechanism of PENG is shown in Figure 2.

The PENG based on a ZnO nanowire array was initially developed by Zhong Lin Wang et al. [59]. In this study, a conductive atomic force microscope tip operating in contact mode was used to deflect vertically oriented nanowires. Electrical current is produced as a result of the Schottky barrier established between the metal tip and the nanowire having a rectifying property. The continuous direct-current nanogenerator was created by the same team in 2007 [60] and is driven by an ultrasonic wave. Zinc oxide nanowire arrays that were vertically aligned and positioned beneath a zigzag metal electrode with a small gap were used to construct the nanogenerator. In order to achieve a number of advantages over generators based on vertically aligned nanowire arrays in terms of stability, robustness, cost, manufacturability, and its ability to work in fluid and under harsh conditions, Rusen Yang et al. [61] further developed a nanogenerator with laterally packaged piezoelectric fine ZnO wires. In addition, the output voltage was estimated to be 15 to 100 times greater than that of fiber and d.c. nanogenerators. Recently, Li-doped ZnO-nanosheets-based direct current high-performance PENGs were demonstrated by Jeongeun Kim et al. [62]. By reducing the number of free electrons in ZnO through doping with the p-type dopant Li, the screening effect was minimized, the piezoelectric output performance was improved, and an output power of 6.55 mW/cm^2^ was attained. One of the main advantages of piezoelectric nanogenerators is that they can convert mechanical energy from various sources, such as human motion, wind, or waves, into electrical energy, which can be used to power small electronic devices or can be stored in a battery for later use [63,64,65]. Additionally, piezoelectric nanogenerators are small in size, flexible, and have low power requirements, which are suitable for a wide range of applications such as wearable devices, self-powered sensors, and energy harvesting systems [66,67,68].

#### Challenges

Despite the many advantages of piezoelectric nanogenerators, there are also several challenges that need to be addressed to make these devices more practical and widely adopted [69]. One of the major challenges of piezoelectric nanogenerators is their low power output, which is usually in the range of micro-watts to milli-watts [70]. Improving the efficiency of nanogenerators and increasing their power output is essential to make them more practical and useful for various applications [71]. Moreover, the mechanical and electrical properties of piezoelectric materials can degrade over time, which can lead to a decrease in the performance of the nanogenerator [72]. Enhancing the durability of piezoelectric materials and the overall design of nanogenerators is important for long-term usage [73]. Most importantly, the costs of piezoelectric materials and fabrication processes are still relatively high, which can make it difficult to produce affordable nanogenerators on a large scale for commercial applications [74,75].

Integrating piezoelectric nanogenerators into various devices and systems is still challenging due to the need for complicated design and integration with other components [76]. Prominently, the environmental factors of temperature, humidity, and pressure can affect the performance of piezoelectric nanogenerators, and this must be taken into consideration when designing and deploying these devices in various applications [57,77]. With these challenges, research works are progressing on improving the performance and practicality of piezoelectric nanogenerators, and it is likely that these devices will play an increasingly important role in sustainable energy generation and harvesting in the future [78,79,80].

### 2.3. Working Principle of Triboelectric Nanogenerators

The function of a TENG is based on triboelectric and electrostatic induction effects, which are a phenomenon in which electrical charges are generated when two materials come into contact and then separate [81]. TENGs operate in four basic modes, as shown in Figure 3: contact and separation, sliding, single electrode, and freestanding. TENGs require the use of two triboelectric materials with proper electrode connections and insulation between each layer. The configuration can be either dielectric–dielectric or metal–dielectric. Furthermore, the choice of dielectric material has the potential to alter the dielectric constant, which is a measure of a material’s ability to store electrical energy. When a mechanical force, such as pressure, bending, or vibration, is applied to the TENG, it causes the two triboelectric layers to come into contact and then separate. When there is displacement in the triboelectric layers, the electrostatic charge movements break the electrostatic status present, and a potential difference between the electrodes develops. The repeated mechanical actuation of layers in the forward and reverse directions causes the triboelectric layer to generate forward and reverse potential between the electrodes, resulting in an electrical charge, as shown in Figure 4. This electrical charge is collected by the electrodes and can be used as a source of energy [82,83].

TENGs are simple and low-cost to produce and can be used to convert mechanical energy from various sources into electrical energy [84]. Moreover, TENGs are highly efficient and can generate high output power compared to other types of energy harvesting devices [85]. They are also small in size and flexible, which leads to an increase in suitability for a wide range of applications such as wearable devices, self-powered sensors, and energy harvesting systems [86,87]. 

#### Challenges

There are several challenges that need to be addressed in order to make TENGs more practical and efficient and reach their potential as a new generation of sustainable energy sources [88,89]. Currently, the energy conversion efficiency of TENGs is still low compared to other traditional energy sources such as solar panels or batteries, which limits the practical application of TENGs [90]. They are often made from delicate materials which can be easily damaged. This results in a short lifespan for TENGs and limits their use in demanding environments [91]. Presently, the production of TENGs is still relatively expensive, which is the main reason they are less accessible to the general public [92]. Moreover, it is difficult to scale up the production of TENGs to meet the energy needs of larger applications due to limitations in size [93]. TENGs are highly sensitive to environmental conditions such as temperature, humidity, and atmospheric pressure. So, it is difficult to use TENGs in extreme environments [94,95]. 

## 3. Materials Used in Nanogenerators

The materials used in nanogenerators play a crucial role in determining the performance and efficiency of the device [96]. In this nanogenerators, some of the most commonly used materials are piezoelectric materials, triboelectric materials, semiconductors, piezoresistive materials, electroactive polymers, and magnetic materials [97]. Piezoelectric materials are materials that can generate an electrical voltage when subjected to mechanical stress [98]. The piezoelectric materials in nanogenerators are PZT, ZnO, and polyvinylidene fluoride (PVDF) [99]. Triboelectric materials are those that tend to lose or gain electrons when they come into contact or separate. Semiconductors are materials that have electrical conductivity between conductors and insulators including silicon, germanium, and graphene compounds. Piezoresistive materials are materials that change their electrical resistance in response to mechanical stress, falling under the category of semiconductors in nanogenerators [100,101]. Additionally, piezoresistive materials in nanogenerators include polymers, carbon nanotubes, and graphene [102].

Alternatively, electroactive polymers are polymeric materials that can change their shape or volume in response to an applied electrical field [103]. The electroactive polymers used in nanogenerators applications are PVDF, polyvinylidene fluoride-co-hexafluoropropylene (PVDF-HFP), polyvinylidenefluoride-co-trifluoroethylene (PVDF-TrFE), polyurethane (PU), polyvinyl chloride (PVC), polydimethylsiloxane (PDMS), and polyethylene terephthalate (PET) [104]. There are also magnetic materials that exhibit magnetostrictive properties that can be used in nanogenerators to convert magnetic energy into electrical energy [105]. The selection of materials for nanogenerators depends on the specific requirements of the device, including the type of energy to be harvested, the desired output power, and the working conditions [45].

## 4. MXenes for Nanogenerators

### 4.1. MXenes

MXenes are a new class of two-dimensional (2D) materials composed of transition metal carbides, nitrides, or carbonitrides [106]. The term “MXene” is derived from the chemical formula of the precursor material, which typically consists of a transition metal (M), a group A element (such as Al or Si), and carbon/nitrogen (X) [107]. The resulting MXene material has a layered structure with a general formula of Mn + 1XnTx, where n represents the number of layers, T represents surface functional groups, and n + 1 is the number of metal layers [108]. MXene can be synthesized from layered ceramic precursors through a process known as etching [109]. MXene materials can be derived from a variety of precursor materials, and the resulting properties of the MXene depend on the choice of precursor and synthesis methods [110].

MXene materials have several unique properties which are promising for use in a variety of applications, including energy storage and conversion, electronic devices, and sensors [111]. There are key properties of MXene materials that are attractive for energy-related applications, including their high electrical conductivity, high surface area, and good thermal stability [112]. These properties make MXene materials ideal for use in energy storage devices such as supercapacitors, as well as energy conversion devices such as thermoelectric generators, solar cells, and batteries [15,113]. In addition to energy applications, MXene materials have also been shown to have potential for use in other areas, including water purification, electromagnetic interference (EMI) shielding, and as catalysts in chemical reactions [114,115]. Despite their promising potential, there are also several challenges associated with the use of MXene materials in practical applications [116]. For example, the synthesis of high-quality MXene materials is still a relatively new field, and further research is needed to fully understand their properties and optimize their performance [117]. Additionally, the relatively high cost of synthesis and processing of MXene materials remains a barrier to widespread commercialization [118]. Conclusively, MXene materials are a promising new class of materials with potential for use in a wide range of energy-related and other applications [119]. Further research and development in this field is likely to lead to new and innovative applications for MXene materials in the future [120].

### 4.2. Different Structures of MXenes

MXenes are typically synthesized by selectively etching the A-layer from MAX phases, which are ternary carbides or nitrides with the general formula Mn + 1AXn, where M is a transition metal, A is an A-group element, X is carbon or nitrogen, and n = 1, 2, or 3 [121]. MXenes can have different shapes depending on the synthesis method used. MXenes are characteristically synthesized as thin nanosheets that are a few nanometers thick and several micrometers wide. These nanosheets have a large surface area and are often used in applications such as energy storage, catalysis, and biosensing [109]. On the other hand, MXenes can also be synthesized as hollow nanotubes with a diameter of a few nanometers to several micrometers. These nanotubes have unique electronic and optical properties and are being explored for applications in electronics, photonics, and biomedicine [122]. MXenes as microspheres with a diameter of a few micrometers to several millimeters are also synthesized. These microspheres have a large surface area and are being explored for applications in energy storage and catalysis [123]. MXenes can be formed as thin films by depositing MXene nanosheets on a substrate. These films have excellent electrical conductivity and are being explored for applications in electronics and energy storage [124]. Based on the structures, aerogels, foams, and sponges are grouped as 3D structured MXenes. These 3D structures have a high surface area and are being explored for applications in energy storage, catalysis, and water treatment [125].

### 4.3. Synthesis of MXenes

The synthesis of MXenes typically involves these preparations of the precursor: exfoliation of the precursor and surface functionalization steps [124]. The precursor is usually a layered transition metal carbide or nitride compound, such as Ti_3_C_2_ or Ti_2_C, that can be synthesized using various methods, including solid-state reactions and chemical vapor deposition [126,127]. The precursor material is exfoliated to obtain individual 2D layers by using a variety of methods including mechanical exfoliation, liquid exfoliation, and electrochemical exfoliation [128]. The surface of the MXene is functionalized with different functional groups, such as carboxyl or hydroxyl groups, to modify its properties and enhance its performance in various applications [129]. The synthesis of MXenes requires a combination of materials science and chemistry knowledge, and the conditions used can significantly impact the quality and properties of the final product. Hence, the careful control of the synthesis conditions and optimization of the process are critical for the successful synthesis of MXenes [118,130,131]. Figure 5 shows the synthesis procedure of MXenes.

The initial progress on the synthesis of various MXenes is highly appreciable as there were a lot of complications involved in that, including the processing of precursors, reaction conditions, and post-processing. Ti_3_AlCl_2_ was derived from Ti_2_Al Cl through a balling process for utilization as a precursor material in the exfoliation process with a hydrogen fluoride (HF) solution and successfully demonstrated the synthesis of two-dimensional Ti_3_C_2_ nanosheets, multilayer structures, and conical scrolls [133]. The low-cost synthesis process of Ti_3_C_2_T_z_ MXene from low-cost precursors such as recycled carbon recovered from waste tires, recycled aluminum scrap, and titanium oxide was demonstrated through the “Evaporated-Nitrogen” Minimally Intensive Layer Delamination (EN-MILD) synthesis approach, and the final MXenes showed better electronic conductivity [134,135]. Including titanium, various transition metal carbides have been attempted through the selective etching process along with different delamination methods, demonstrating successful synthesis of a two-dimensional structure with proper surface termination [136]. The chemical vapor deposition (CVD) process was also attempted for the development of transition metal carbides, and a successful demonstration for the synthesis of molybdenum carbide (α-Mo_2_C) has also been reported [137,138]. Salt–acid etching followed by a sonication process were demonstrated for the successful synthesis of a vanadium nitride MXene [139]. A high-energy ultrasonic cell-crushing extraction method to successfully prepare Ti_3_C_2_T_x_ MXenes from Si-based MAX using a single low-concentration etchant was demonstrated with a fast processing time [140]. Various etching processes including HF etching, in situ HF-forming etching, electrochemical etching, alkali etching, and molten salt etching, along with delamination strategies with proper demonstration, were reported [141]. Scalable synthesis, a fundamental process involved in synthesis, is the role of various metal ions in the synthesis of MXene, and safety guidelines to reduce the risk during the synthesis process of MXene have also been reported [142,143,144]. With this knowledge, further processes are still ongoing to reduce the cost, time, and risk. 

### 4.4. Properties of MXenes

The basic building block of MXenes is a metal–carbon or metal–nitrogen layer, where the metal atoms are surrounded by carbide or nitride anions. The metal–carbon or metal–nitrogen layer is sandwiched between two graphene-like layers. The graphene-like layers are composed of hexagonal arrays of metal and anion atoms [145,146,147]. The metal atoms in the MXene structure have coordination numbers ranging from 4 to 6, and they bond covalently with the anion atoms [148]. The interlayer spacing between the metal–carbon or metal–nitrogen layers is very small, typically less than 0.5 nm [149]. This gives MXenes their unique combination of high mechanical strength and electrical conductivity [150]. The overall structure of MXenes is highly ordered, with well-defined layers that can be easily exfoliated to produce single- or few-layer materials [151]. This makes MXenes highly suitable for different applications including energy storage, catalysis, and electronics [152].

MXenes have several unique properties that make them highly attractive for use in a wide range of applications, including energy storage and conversion, electronic devices, and sensors [153]. MXenes have high electrical conductivity, making them ideal for use in applications such as energy storage devices and electronics [154]. Moreover, they have a high surface-area-to-volume ratio, which is important for applications such as energy storage devices, catalysts, and sensors [155]. MXenes have good thermal stability, which is important for applications such as energy storage and conversion devices where they are subjected to high temperatures [156]. The mechanical properties of the devices should be appropriate after the fabrication. MXenes have excellent mechanical properties, including high strength, flexibility, and durability, and that is why they are attractive for use in flexible electronics and wearable devices [157]. MXenes have good chemical stability, which is important for applications such as water purification and catalysts [158]. MXenes have been shown to have good biocompatibility, which is highly suitable for use in biomedical applications such as drug delivery and tissue engineering [159]. MXenes have been shown to have high hydrogen storage capacity for use in hydrogen fuel cells [160]. Conclusively, unlike other 2D materials such as graphene, MXenes are possible to synthesize using a relatively low-cost process [121]. The combination of these properties gives special attention to MXenes for use in a wide range of energy-related and other applications [161].

MXenes have several mechanical characteristics, which is highly suitable for energy harvesting applications [162]. MXenes have a high Young’s modulus, which indicates their ability to resist deformation under stress with high tensile strength, which resembles those properties resisting fracture under tensile condition [163]. Moreover, MXenes possesses high fracture toughness, which is important for resisting crack propagation [164]. Based on the tribological behavior, MXenes have a low coefficient of friction and good wear resistance, which contribute more in tribological applications [165]. Apart from this, some MXenes such as Ti_3_C_2_Tx can be fabricated into flexible films, which results in using the flexible films in flexible electronics and energy storage devices [166]. The mechanical properties of MXenes are to be fine-tuned by controlling their composition, synthesis method, and processing conditions [167].

The structure of MXenes typically consists of a transition metal layer sandwiched between two surface functional groups such as hydroxyl or fluorine [168]. These surface functional groups allow for easy exfoliation of the MXene layers and also provide opportunities for chemical modification [117]. The interlayer spacing in MXenes can be controlled by varying the size and nature of the surface functional groups [169]. 

## 5. MXene /Polymeric Composites for Energy Harvesting 

### 5.1. Fabrication of MXene Polymer Composite

The fabrication of MXene–polymer composites involves combining MXene 2D materials with a polymer matrix to create a new class of materials with improved properties [170]. The first step of the fabrication process is preparation of MXene suspensions. The MXene flakes are dispersed in a solvent to form a homogeneous suspension which can be stabilized using surfactants or stabilizing agents [171]. Then, the polymer matrix is usually prepared by melt mixing or solution casting, depending on the compatibility of the polymer with the solvent used for the MXene suspension. Afterwards, the MXene suspension is added to the polymer matrix to form a composite material. The mixing process can be performed using various methods such as manual mixing, sonication, or mechanical stirring [172]. Finally, the composite material is then dried to remove the solvent, and the remaining polymer matrix is consolidated through heating or mechanical pressure to form a dense composite material [173]. Figure 6 depicts the preparation process of MXene–polymer composites as well as various methods of preparation such as spin coating, electrospinning, and hot press techniques.

The MXene–polymer composite is then characterized using various techniques such as transmission electron microscopy, scanning electron microscopy, X-ray diffraction, and thermogravimetric analysis to evaluate its structure, composition, and properties [174,175]. The fabrication of MXene–polymer composites can be optimized by controlling the mixing conditions, the concentration of MXene suspensions, and the type of polymer matrix used, among other factors involving fabrication. This allows for the tailoring of the properties of the composite material to meet the requirements of specific needs [176,177,178,179]. A modified Ti_3_C_2_Tx MXene (m-MXene)-based nanocomposite elastomer-based ultraflexible and self-healing TENG was designed by Yuzhang Du et al. [180]. The single-electrode mode-based TENG can produce a high open-circuit voltage (Voc) of 245 V and a high short-circuit current (Isc) of 29 A, because of the exceptional electronegativity of m-MXene. By taking advantage of the functional groups present in MXene, S M Sohal Rana et al. [181] developed an electrospun nanofiber-based TENG (EN-TENG) using a PVDF-TrFE/MXene nanocomposite material with a superior dielectric constant and high surface charge density. A dipole can be created by the H bonds between the H atoms on the polymer chains and the O and F atoms, which are negative charges on the MXene surface. The main element for improving the dielectric constant of the PVDF-TrFE/MXene composite is large dipole moments of hydrogen bonds. Trilochan Bhatta et al. [182] fabricated electrospun fibers by incorporating MXene nanosheets into the PVDF matrix, taking advantage of the high dielectric property and surface charge density of MXene. This significantly enhanced the triboelectric performance of the nanofiber. When conductive MXene nanosheets were incorporated into PVDF nanofibers, the dielectric constant and surface charge density of the nanofibers were considerably increased by 270% and 80%, respectively. Stretchable TENG was developed by Sujoy Kumar Ghosh et al. [183] using 2D MXene nanosheets and ferroelectric BTO. The dielectric constant rises when MXene is used, and the dielectric loss is decreased by coupling with the ferroelectricity of BTO, improving the nanogenerator’s total output performance. Because of its extremely negative triboelectric characteristics and mechanical stability, Md. Salauddin et al. [184] designed a unique MXene/Ecoflex nanocomposite that is a potential triboelectric material by utilizing the strong mechanical stability and electronegativity of MXene. The fabricated TENG is used to harvest energy from biomechanical motions, wind, and rain. A triboelectric and piezoelectric hybridization generator was constructed by Jonghyeon Yun et al. [185] by incorporating MXene and barium titanate ceramic filler in the PDMS matrix. The function of MXene as a bifunctional conductive filler with the ability to induce and trap charges was demonstrated. A high Voc of 80 V, an Isc of 14 A, and a power density of 13.5 W/m^2^ are achieved when the MXene concentration is examined at its optimum. From the above discussion, we could understand that different parameters were involved in the enhancement of the performance of the nanogenerators. The next section will discuss the fabrication and performance of the MXene with PVDF as a composite.

### 5.2. MXene/PVDF Composites

PVDF is a semi-crystalline polymer which exists in two different phases, the crystalline phase (β–phase) and the amorphous phase (α–phase), as shown in Figure 7. The proportions of these two phases determine the properties of the PVDF material [186,187]. The first one is the crystalline phase, which is highly ordered with aligned dipoles [188]. This phase is responsible for the high electroactive nature, high dielectric properties, mechanical strength, thermal stability, and electrical insulation properties of PVDF, which makes it useful for applications that require polar behavior, such as piezoelectric sensors and energy harvesting devices [189]. The second one is an amorphous phase in which the dipoles are not aligned due to random arrangement of the polymer chains [190,191]. The proportions of these two phases are controlled through processing conditions such as temperature, heating and cooling rate, and strain, which can affect the crystallinity of the PVDF material [192]. Evidently, the rapid cooling during processing results in a higher proportion of the crystalline phase, while slow cooling can result in a higher proportion of the amorphous phase [193].

PVDF is widely used due to its unique combination of physical and chemical properties. Moreover, the dielectric properties of PVDF are a critical factor in its use as a dielectric material in electrical and electronic applications [194,195,196]. PVDF has a high dielectric constant, meaning it can store a large amount of electrical energy per unit volume. This makes it useful for energy storage applications, where it can be used as a dielectric material in capacitors [197,198]. Then, PVDF has high dielectric strength, meaning it can withstand high electrical fields without breaking down or becoming electrically conductive [199]. It has low dielectric loss, which loses only a small amount of electrical energy as heat during energy storage [200]. This leads to usefulness for energy storage applications, where it can be used as a dielectric material in capacitors [201,202]. 

The composites of MXene and PVDF materials can be used for energy harvesting applications including triboelectric and piezoelectric nanogenerators. The PENGs are working completely based on the piezoelectric effect, which is used to generate electrical energy. The combination of MXene and PVDF in PENGs can result in higher electrical output and improved energy conversion efficiency compared to traditional PVDF-based PENGs [203,204,205]. The potential benefits of using MXene–PVDF composites in energy harvesting applications include their potential for new applications in wearable electronics and the Internet of Things (IoT) as well as their potential for use in harsh environments. However, there are still some challenges associated with the use of MXene–PVDF composites in energy harvesting, including the synthesis and processing of high-quality MXene materials and the optimization of the interface between the MXene and PVDF materials [24,117,206]. 

Similar to the conductive reinforcements represented by graphene and carbon nanotubes, MXene can also stimulate piezoelectric phase formation, induce a strong interfacial coupling effect, and yield an enhanced piezoelectric response in PVDF. Moreover, it should be noted that the reinforcing effect induced by an MXene nanosheet is more significant as compared with graphene and carbon nanotubes upon a similar loading level. For example, the reported maximum piezoelectric coefficient d_33_ of carbon nanotubes/PVDF is 23 pC/N at a loading of 0.2 wt%, while that of MXene/PVDF can reach 35 pC/N at the same loading [207]. Figure 8a,b shows a scanning electron microscope (SEM) image of a multilayered MXene after the etching MAX phase and a clear transmission electron microscope (TEM) image of MXene nanosheets after delamination processes, respectively. Upon a loading of 0.4 wt%, the piezoelectric coefficient d_33_ of the MXene/PVDF hybrid film reaches a peak of 43 pC/N. Meanwhile, the incorporation of MXene nanosheets is found to strengthen the mechanical performance and obtained a Young’s modulus of 1.49 GPa, as shown in Figure 8c,d. With an excellent piezoelectric response and mechanical property, the MXene/PVDF hybrid film sensor demonstrates an ability to perform superior voltage response under a load of 150 N and voltage sensitivity up to 0.0480 V/N, which is double that of a PVDF-based sensor (Figure 8e,f). Accompanied with excellent sensitivity, the hybrid film sensor also demonstrates an ability to operate stably under cyclic strains, suggesting its applicability as reliable pressure sensors [208]. 

Jinyoung Kim et al. [209] demonstrated self-powered piezoelectric e-skins with good sensitivity and a wide detection range using 3D porous MXene/PVDF structures. To enhance the ferroelectric properties of PVDF, MXene was used as a nucleation agent, and a piezoelectric e-skin with 3D porous microstructures and Ni/Cu electrodes on both sides, made of MXene/PVDF composite, is schematically represented in Figure 9(Aa). The considerable surface functional groups of MXene, including oxide –O, –OH, and –F, could initiate intermolecular hydrogen bonding with –CH_2_ and –CF_2_ groups of PVDF, leading to the intercalation and confinement of PVDF among MXene nanosheets and the crystallization of the highly polar β-phase PVDF. In addition, MXene nanosheets could serve as nucleation agents to enhance the electroactive properties of PVDF effectively, in contrast to spherical or nanorod shapes, due to their large specific area in 2D geometry (Figure 9(Ab)). Figure 9(Ac) shows the piezoelectric output current. The SEM image and photograph of the porous MXene/PVDF e-skin are shown in Figure 9(Ba,b). The porous piezoelectric e-skins exhibited piezoelectric sensitivities of 11.9 and 1.4 nA/kPa in the low (2.5 kPa) and high (2.5–100 kPa) pressure ranges, respectively, which were 31 and 3.7 times higher compared to the planar MXene/PVDF e-skins’ sensitivity of 0.4 nA/kPa. To monitor the radial artery pulse, the porous MXene/PVDF e-skin was affixed to the wrist and continuously measured the output values in real-time (Figure 9C). The measured pulse frequency, which was 68 beats per minute, was indicative of a stable state and consistent with that of healthy adults (Figure 9C). E-skins with rapid piezoelectric response enabled spatial and temporal detection of vibrations generated during scanning, thus enhancing the perception of surface textures. 

The schematic diagram illustrating texture perception using an e-skin is shown in Figure 9(Da). To amplify the piezoelectric output triggered by surface texture, they utilized line-patterned PDMS (Figure 9D), inspired by the amplification of tactile signals by fingerprint patterns. The time-dependent piezoelectric current output values were measured as the porous MXene/PVDF e-skin was scanned over the PDMS surface as shown in Figure 9(Db). As the scanning speed increased, both the intensity and number of periodic spike peaks exhibited a corresponding increase.

Guo Tian et al. [210] studied a dielectric micro-capacitance-enhanced piezoelectricity strategy that involves texturing well-aligned MXene (Ti_3_C_2_T_x_) sheets within a PVDF matrix using a scalable blade-coating technique. By introducing interfacial polarization, the dielectric micro-capacitance increases the permittivity, resulting in a significant improvement in piezoelectricity. The aligned MXene/PVDF composite produced an exceptional piezoelectric coefficient of 63.3 pC/N. Additionally, a modified piezo-composite model is suggested to explore the underlying mechanism of interfacial polarization and dielectric relaxation effects in controlling piezoelectricity, potentially aiding in the development of guidelines for designing piezo-composites. This study reveals a fresh perspective on comprehending the electrical properties of polymer composites and opens up avenues to fabricate innovative composites with exceptional piezoelectricity.

Flexible piezoelectric films were prepared via electrospinning by incorporating MXene into PVDF-TrFE, and the influence of MXene on the mechanical and piezoelectric properties of the composite films was investigated. From Fourier-transform infrared spectroscopy (FTIR), studies as shown in Figure 10(Aa), no α-phases are present in the composites due to the electrostatic interaction between MXene and PVDF-TrFE (Figure 10(Ab)). Compared to neat PVDF-TrFE nanofibrous films, the PVDF-TrFE/MXene nanofibrous films prepared in this study produced significantly stronger electrical signals under the same pressure conditions. For instance, the composite films containing 2.0 wt% MXene generated a V_OC_ and power density of 1.5 V and 3.64 mW/m^2^, respectively, when subjected to a pressure of 20 N at 1 Hz (Figure 10(Ba)). In addition, the composite films exhibited a desirable linear relationship between V_OC_ and pressure, making them suitable for pressure-sensing applications (Figure 10(Ac)). These films can also harness energy by transforming mechanical energy into electrical energy under various motions such as wrist bending, finger touching, and foot sweating (Figure 10(Bc–f)), thus making them ideal for creating self-powered piezoelectric pressure sensors [211].

Triboelectric nanogenerators based on MXene@PVDF are a type of energy harvesting device that generates electrical energy through the interaction of two different materials. A thin layer of MXene material is deposited on a PVDF film in this type of TENG, resulting in a triboelectric nanogenerator with improved performance when compared to traditional PVDF-based TENGs. The advantage of using MXene in TENGs is that it matches its high electrical conductivity and good mechanical properties. These combined properties allow the MXene@PVDF TENG to produce higher electrical output and improved energy conversion efficiency compared to bare PVDF-based TENGs [18,19,212,213]. In addition, the integrated MXene with PVDF also offers the potential for TENGs to be used in harsh environments where traditional TENGs may not be applicable to perform successfully. Even with these advantages, there are still some challenges connected with the use of MXene@PVDF TENGs, including the synthesis and processing of high-quality MXene materials and the optimization of the interface between the MXene and PVDF materials. However, the prospective benefits of MXene@PVDF TENGs are making an exciting area of research and development [214,215].

A high-performance biomechanical energy harvesting device is being developed for human finger tapping using an electrospun nanofiber-based TENG with PVDF-TrFE/MXene nanocomposites. The incorporation of MXene nanosheets into the PVDF-TrFE polymer matrix significantly enhances the electronegativity, resulting in a remarkable improvement in the EN-TENG performance. The addition of MXene as a conductive filler in the PVDF-TrFE polymer matrix has considerable effects. It improves surface charge density through its capacitive effect, while reducing surface density by leaking current through its conductive effect. PVDF-TrFE, known for its high electronegativity and chemical and mechanical stability, was selected for the nanofiber mat. By adding MXene, the novel composite material (PVDF-TrFE/MXene) is able to increase the dielectric constant, surface potential, and surface charge in the friction layer of the nanofiber mat. The incorporation of MXene sheets with terminating functional groups along with the −F groups of PVDF-TrFE leads to the formation of microscopic dipoles, thereby increasing the surface charge density. This phenomenon enhances the dielectric constant of the composite materials, which is due to the charge accumulation at the nanofiber surface between the polymer and MXene sheets. Moreover, the high electronegativity of MXene contributes to the improved output performance of the EN-TENG [181].

Viet Anh Cao et al. [216] used a hydrothermal method to chemically modify the electrical polarity of Ti_3_C_2_ nanoflakes to serve as positive or negative fillers in the friction layers of TENGs (Figure 11a). Nanoflakes functionalized with -NH_2_ groups (NH_2_-Ti_3_C_2_) showed triboelectrically positive properties while N-functionalized Ti_3_C_2_ (N-Ti_3_C_2_) exhibited triboelectrically negative characteristics. Composites of chemically modified Ti_3_C_2_ nanoflakes with ferroelectric PVDF-TrFE and Nylon 11 were utilized to fabricate positive and negative friction layers of TENGs, resulting in PVDF-TrFE/N-Ti_3_C_2_ and Nylon 11/NH_2_-Ti_3_C_2_ composite layers, respectively (Figure 11b). The functionalized Ti_3_C_2_ of different polarities was used to modulate the surface potential of each friction layer and to investigate the role of fillers in the polarization of ferroelectric polymers. Furthermore, the incorporation of Ti_3_C_2_ improved the mechanical properties of the friction layers in both matrix polymers. The effect of functionalized Ti_3_C_2_ on TENG output Voc was also studied. The PVDF-TrFE/N-Ti_3_C_2_ and Nylon 11 friction pairs had the highest output open-circuit voltage among the samples, and N-Ti_3_C_2_ was found to be better filler for PVDF-TrFE used as the negative friction layer (Figure 11c). Similarly, the triboelectric outputs of Nylon 11 composite layers mixed with Ti_3_C_2_ or functionalized Ti_3_C_2_ against a PVDF-TrFE negative friction layer were used to determine a suitable filler for Nylon 11 used as a positive friction layer. The highest output open-circuit voltage was obtained in the case of the Nylon 11 layer when it was composited with NH_2_-Ti_3_C_2_ (Figure 11d). The TENGs produced in this study attained a peak open-circuit voltage of 250 V, a short-circuit current density of 280 µA/cm^2^, and an output power density of 13 mW/cm^2^. Moreover, they exhibited steady outputs during 14,400 cycles, indicating remarkable stability and durability. Due to the ferroelectric properties of both composite polymers, hysteresis loops were observed in the polarization–electric field (P-E) curves obtained after polarization (Figure 11d,e). Furthermore, using the output power generated, 50 light-emitting diodes (LEDs) connected in series could be illuminated or used to power a digital hygrometer (Figure 11f). TENGs can also be used to harvest energy and track human movement. TENGs were attached to the shoe insole to demonstrate this, as shown in Figure 11g, to harvest energy from walking and running. Further demonstrations of TENGs’ potential applications as humidity sensors were also shown (Figure 11h). Still, there are more opportunities in the MXene/PVDF-based composites, as we have more experimental parameters to optimize to obtain a better composite for better performance. 

### 5.3. MXene/PDMS Composites for Energy Harvesting

PDMS is a silicone polymer that has several properties which make it useful for energy conversion applications. Based on the energy conversion, the PDMS has a high elastic modulus which makes it highly flexible and able to deform under stress. This property makes it useful for applications of energy harvesting where it can be used as a flexible substrate to convert mechanical energy into electrical energy [217]. Moreover, it has good dielectric properties, meaning it has a high electrical resistance and can store electrical charges. This makes it useful for energy storage applications where it can be used as a dielectric material in capacitors. Meanwhile, it is transparent in the visible and near-infrared regions of the electromagnetic spectrum, making it useful for solar cell applications where it can be used as a protective layer to prevent environmental damage and improve the overall efficiency of the cell [218,219,220,221]. To withstand thermal conditions, PDMS has a wide temperature range in which it retains its properties, which is appropriate for energy conversion applications that require high-temperature stability. Additionally, it is highly resistant to chemicals, including water, oils, and most organic solvents, which makes it useful for results requiring chemical stability [222]. Hence, the properties of PDMS and PDMS composites confirm them as a prospective material for energy conversion applications including energy harvesting, energy storage, solar cells, and fuel cells. The flexibility, dielectric properties, transparency, thermal stability, and chemical resistance of PDMS are the main prominent properties to make appropriate nanogenerators [223,224].

The composites of PDMS and MXene have been studied for their potential in energy harvesting applications. The unique combination of electrical conductivity and mechanical flexibility of PDMS and MXene composites is suitable for a variety of energy harvesting applications such as flexible piezoelectric energy harvesters and thermoelectric generators [185,225]. In these applications, the PDMS provides mechanical flexibility and damping while, on the other hand, the MXene contributes to electrical conductivity. It is important to note that while PDMS–MXene composites have shown promise for energy harvesting applications [226], much attention is still needed for optimizing the performance and scalability of these materials for real-world applications.

Liu et al. [227] developed a multifunctional TENG device based on a single-layer structure of stretchable PDMS/MXenes, which can effectively detect changes in the environment and convert mechanical and light energies into electricity. A metal–insulator–metal (MIM) capacitor model was developed to understand the mechanism of the significant increase in output performance of the PDMS/MXene-composite TENG. The pure PDMS behaves as a typical dielectric polymer in the presence of an externally applied electric field, where charges accumulate near the electrodes. In contrast, with the external electric field, the MIM capacitor that uses the PDMS/MXene composite as a dielectric layer exhibits a different phenomenon, where charges accumulate near the electrodes and the interfaces between the MXenes and PDMS, which differs from the pure PDMS film and offers high output performance. Figure 12a–d demonstrates the output performance of the PDMS/MXene TENG under various mean pressures, light illuminations, and frequencies. Due to the high electronegativity of the PDMS/MXenes film, the hybrid TENG can produce a V_oc_ and I_sc_ of 145 V and 27 μA, respectively. Furthermore, the output performance is enhanced 3.1 times when the light intensity is increased compared to the TENG without light illumination, and the light power conversion efficiency reaches 19.6%. The TENG sensor produced in this study demonstrated remarkable human–computer interaction capabilities such as high stretchability (123%) and exceptional acoustical detectivity. In addition, an environmental interaction visualization system that improves the efficiency and simplicity of social interactions was presented, and a color-tunable LED actuated by our TENG was used to record voltage changes under different environments such as mechanical wave and light irradiation (Figure 12e,f). The multifunctional TENG device has the potential to efficiently harvest energy from the surrounding environment and is highly promising for future wearable electronic applications [227].

Jiang et al. [228] developed an advanced PDMS/MXene composite to harvest electricity from leaf swing energy and handwriting motion energy. The composite film demonstrated enhanced triboelectronegativity as well as increased electrical conductivity in the TENG. Taking advantage of the leaf’s ability to donate electrons, a triboelectronegative PDMS/MXene composite film was adhered to the leaf. Then, on the other side of the PDMS/MXene composite film, a flexible laser-induced graphene (LIG) electrode was placed. All of these components were combined into a TENG that operates in a single-electrode mode (Figure 13(Aa)). The leaves were swung by an electric fan. The wind generated by the fan blows the leaves, producing a significant output voltage instantly. Furthermore, as shown in Figure 13(Ab), the generated signal increases with increasing wind speed, with a maximum voltage of 2 V obtained. They also developed TENG as a writing board to harvest writing energy in daily life. Upon writing on the paper, the electric energy is generated, and when writing is stopped, there is no obvious output signal, as shown in Figure 13(Ac) [228].

For the purpose of developing high-output flexible, washable, and durable single-electrode triboelectric nanogenerators (STENGs), a PDMS/MXene composite film is presented with improved power output (Figure 13(Bb)). It is believed that this is the first time such a composite film has been demonstrated. The presence of Ti, C, and F elements across the scan area in the EDX mapping of composite film confirmed the uniform distribution of MXene in PDMS (Figure 13(Ba)). The flexible STENG showed remarkable endurance against severe deformation (folding and rolling) and remains waterproof even after washing (Figure 13(Bc)). Furthermore, a hand hammering test was conducted to validate the practical application of the STENG converting the external mechanical energy from human motion into electrical energy (Figure 13(Be)). The flexible STENG developed in this study was able to directly power 80 commercial green LEDs without additional energy storage devices by harnessing energy from continuous human hand hammering motion. Furthermore, the device was tested for practical applications and was attached to a human finger and leg, where it was able to detect finger tapping, hand clapping, and hand hammering as a human motion sensor. This innovative design provides an appealing concept for various wearable electronics and flexible self-powered systems [229]. 

Peida Li et al. [230] fabricated a soft actuator that can be performed for moisture-driven actuating, humidity energy harvesting, self-powered humidity sensing, and real-time motion tracking, all in one device. Unlike other humidity-responsive actuators, this actuator can harvest humidity energy and convert it into both mechanical power and electricity to enable self-powered sensing of humidity and motion modes in real-time. The actuator is made of a composite membrane (MCPM) of MXene, cellulose, and polystyrene sulfonic acid (PSSA), supported by polyethylene terephthalate (PET). Directional proton diffusion and asymmetric expansion under moisture gradient are used to integrate these multiple functionalities into the MCPM. The soft actuator obtained in this study demonstrates a maximum bending angle of 130 degrees, accompanied by a maximum power density of 81.2 μW/cm^3^ and an open-circuit voltage of 0.3 V, which makes it an ideal candidate for developing power sources, self-powered smart switches, soft robots, and human touch and breathing monitors with high sensitivity. 

Moreover, the researchers successfully fabricated highly flexible PDMS nanocomposites containing 3D interconnected MXene using the unidirectional freeze-drying and vacuum-assisted impregnation techniques [231]. The schematic of the preparation process is shown in Figure 14(Aa). The SEM images of the cross-section, as shown in Figure 14(Ab,c), revealed a typical hierarchical architecture of 3D MXene aerogels, indicating that the MXene flakes were completely dispersed in the aqueous solution due to their high hydrophilic nature. The pore size of MXene aerogels appeared to decrease as MXene concentration increased to 2.5 vol%. Meanwhile, in these aerogels, MXene flakes were interconnected horizontally and vertically stacked one by one to self-assemble a 3D MXene skeleton. This 3D network has the potential to improve charge carrier and heat transfer efficiency. The micro-pores of 3D MXene aerogels were fully filled with PDMS after impregnation, as shown in Figure 14(Ae,f). The composites show no apparent interface cracks or phase separation between the 3D MXene skeleton and the PDMS matrix, indicating good interfacial adhesion. 

These nanocomposites demonstrate high thermal and electrical conductivities. The MXene nanosheets are connected to each other, forming a 3D network within the PDMS matrix. In comparison to neat PDMS, the 3D MXene/PDMS composites demonstrated excellent electrical conductivity of 5.5 S/cm, which was 14 times higher than PDMS, and thermal conductivity enhancement of approximately 220% at a low MXene content of 2.5 vol% (Figure 14B). Moreover, the 3D MXene/PDMS composites can also serve as a negative tribo-material in TENG, resulting in an increased output current of the device (Figure 14C). These findings suggest that the 3D MXene/PDMS nanocomposites have great potential for various applications in flexible nanogenerators and thermal management [231].

On the other hand, the use of a low-cost lotus-leaf-bionic (LLB) method instead of photolithography to create microstructures on the surface of the PDMS negative triboelectric layer has been demonstrated by Wang et al. [232] to produce high-output TENGs. To evaluate the TENGs, the PDMS film was modified by doping 2D MXene (Ti_3_C_2_Tx) and graphene. The MXene-doped LLB TENG showed a maximum output power density of 104.87 W/m^2^, which was approximately 10 times higher than that of the graphene-doped LLB TENG. The MXene-doped LLB TENG produced in this study can serve as a humidity sensor with high sensitivity of 4.4 V per RH% and can power low-energy electronic devices such as electronic watches and green LED arrays. Moreover, the MXene-doped LLB TENG can harvest energy from various human motions, making it an excellent candidate for constructing a health monitoring system for human feet, which was demonstrated for the first time in this study. These findings highlight the potential of low-cost TENGs produced by bionic means and their promising applications in smart human health monitoring systems. 

### 5.4. MXene/PVA Composites for Energy Harvesting

PVA (Polyvinyl Alcohol) is a synthetic polymer that is water-soluble and can be used in a variety of applications. It is produced by the polymerization of vinyl acetate monomers, which are then partially or fully hydrolyzed to form PVA. PVA is a versatile material with a wide range of properties, including high tensile strength, flexibility, and resistance to grease, oil, and solvents. It is also biodegradable, making it a more environmentally friendly alternative to many other synthetic polymers. The main advantages of PVA are its water solubility, biodegradability, and versatility, which make it a useful and sustainable option for many applications in different industries. The molecular structure of PVA is composed of repeating units of vinyl alcohol monomers, which have the chemical formula CH_2_CHOH. These monomers are linked together through ester linkages (COO) to form a polymer chain. The repeating unit of PVA can be represented as (-CH_2_CHOH-)n where “n” represents the number of repeating units in the polymer chain. The hydroxyl (-OH) groups on the polymer chain give PVA its water-soluble properties. Depending on the degree of hydrolysis, some or all of the acetate groups (-OCOCH_3_) on the polymer chain may be replaced by hydroxyl groups (-OH). This affects the properties of PVA, such as its degree of crystallinity, mechanical strength, and water solubility. When PVA and MXene are combined, the resulting composite exhibits a range of improved properties, such as increased tensile strength, improved thermal stability, and enhanced electrical conductivity. These properties make PVA/MXene composites potentially useful in a variety of applications, including energy storage devices, sensors, and electromagnetic interference shielding. 

A TENG with multiple functions was fabricated by using a combination of PVA/silver (Ag) nanofibers and fluorinated ethylene propylene (FEP) film for monitoring human respiration, movement, and harmful gases. The electrospun PVA/Ag nanofiber film, which serves as a high-performance triboelectric material, significantly enhanced the TENG’s output performance. To generate electricity from wind energy and monitor the environment simultaneously, TENGs based on PVA/Ag nanofibers and FEP films were developed. The electrospun PVA/Ag nanofiber film, as an effective triboelectric material, significantly improves the output performance of the TENG, which can achieve an output voltage and power density of up to 530 V and 359 mW/m^2^, respectively, at a wind speed of 8 m/s. The self-powered sensor can operate independently without an external power source, and its output voltage (36 V) can be regulated at a wide range of wind speeds (2.5–8 m/s) and relative humidity (30–65% RH) by using the voltage regulator module. The TENG-driven Ti_3_C_2_T_x_ MXene/WO_3_-based NO_2_ sensor exhibits outstanding sensitivity with a response of ΔUs/Usa = 510% @ 50 ppm at room temperature, which is 15 times higher than that of the resistive sensor. To detect the wind direction and track the origin of harmful gases, a multifunctional detection system comprising four TENGs and a gas sensor was created [233].

Xiongxin Luo et al. [234] successfully fabricated triboelectric nanogenerators that possess flexibility and stretchability, utilizing MXene/PVA hydrogel as the electrode material. The inclusion of MXene nanosheets in the PVA hydrogel not only facilitated crosslinking but also created a microchannel within the hydrogel. These microchannels contributed to the composite hydrogel’s stretchability, improved ion transport, and additionally generated additional output through the SVP mechanism of microchannel triboelectricity. It is shown that an ideal doping concentration of 4% MXene nanosheets was optimal for MXene/polyvinyl alcohol (PVA) hydrogel TENGs (MH-TENGs). These MH-TENGs exhibit exceptional stretchability and remarkable sensitivity to mechanical stimuli, highlighting their immense potential for applications such as wearable self-powered body movement monitoring and high-precision written stroke recognition. Moreover, the freestanding triboelectric materials used in MH-TENGs can be seamlessly integrated with diverse materials, showcasing immense potential in harnessing low-frequency mechanical energy. Importantly, the MH-TENG exhibits degradability and environmental friendliness, aligning with the current emphasis on environmental preservation. The Ti_3_C_2_Tx MXene structure exhibits an alternating arrangement of Ti and C atoms, following the sequence of Ti/C/Ti/C/Ti. Additionally, the terminal functional groups (represented as Tx) are distributed on the surface of the layered MXene as illustrated in Figure 15(Aa). The TEM images of layered MXene are shown in Figure 15(Ab,c). The energy-dispersive X-ray (EDX) spectroscopic mapping (Figure 15(Ae)) demonstrates the evenly distributed Ti, F, O, and C elements across the MXene nanosheets. Hence, the MXene/PVA hydrogel, as fabricated, exhibited remarkable self-healing capability (Figure 15(Af)), allowing it to quickly and seamlessly heal when the divided parts were rejoined in the natural environment. Moreover, Figure 15(Ba) presents the schematic representation of the synthesized MXene/PVA hydrogel structure. In the synthesis process, borate molecules were employed as crosslinking agents to enhance the properties of the pure PVA hydrogel. The Raman spectra (Figure 15(Bc)) of both the MXene nanosheets and the hydrogels exhibit similarity to a previous report, indicating consistency in their structural properties. X-ray diffraction (XRD) (Figure 15(Bd)) analysis of the MXene nanosheets revealed distinct peaks associated with the (002), (006), and (008) planes, whereas the MXene/PVA hydrogel exhibited the absence of characteristic peaks. This observation suggests that the MXene/PVA hydrogel retained its cross-linked state (Figure 15(Ba)). The developed MXene/PVA hydrogel was experimented with using different types of surfaces, and their recorded responses are shown in Figure 15(Ca–f). Moreover, the different handwritten styles have been recorded, as shown in Figure 15(Da–f). 

Dongyue Wang et al. [235] introduced a self-powered humidity-sensing device that utilizes a monolayer MoSe_2_ PENG for accurate humidity detection. The fabrication of PENG with MoSe_2_ is shown in Figure 16(Aa). Moreover, the PVA/MXene is fabricated by electrospinning, and a humidity sensor was prepared using PVA/MXene (Figure 16(Ab,c)). In Figure 16(Ad–f), the illustration of prepared monolayer MoSe_2_ by the APCVD process and fabricated piezoelectric devices with two electrodes along with a flexible PENG device is shown. Here, the piezoelectric characteristics of monolayer MoSe_2_ were investigated and reported for the first time. Under a strain of 0.36% and a frequency of 0.5 Hz, the PENG demonstrated an impressive peak output of 35 mV. Moreover, the flexible PENG has the ability to harvest energy and produce varying output voltages depending on its placement on different areas of the human body. A self-powered sensor was prepared using a PVA/MXene composite nanofiber film and powered by the monolayer MoSe_2_ PENG. This sensor effectively detects humidity by converting mechanical energy into electrical energy, resulting in a significantly enhanced response (40) that is 40 times higher compared to pure MXene (Figure 16B). Additionally, the humidity sensor exhibits rapid response and recovery times of 0.9/6.3 seconds, minimal hysteresis of 1.8%, and consistent repeatability. Furthermore, the PVA/MXene nanofiber film was employed to fabricate a flexible humidity sensor on a PET flexible substrate, showcasing exceptional capabilities in detecting moisture on the surface of human skin (Figure 16(Cc–f)). Still, a lot of investigations are ongoing in order to achieve further improvement in the device’s performance.

### 5.5. Other MXene Composites for Energy Harvesting

Other than polymeric materials, several materials were attempted to form a composite with MXene, and their performances towards energy harvesting were evaluated. Carbon nanotubes (CNTs) are considered to be one of the strongest and stiffest materials known, as well as one of the most thermally and electrically conductive materials. Moreover, they can be synthesized in a variety of ways including chemical vapor deposition, arc discharge, and laser ablation. Moreover, the properties of CNTs can be modified by controlling their diameter, chirality, and length, as well as by functionalizing their surface with other chemical groups [236,237,238]. The molecular structure of CNTs is based on a rolled-up graphene sheet which is made up of hexagonal arrays of carbon atoms. The carbon atoms in a graphene sheet are bonded covalently in sp2 hybridization, forming strong, planar bonds which give graphene its characteristic mechanical and electronic properties. In a CNT, the graphene sheet is rolled up into a cylindrical shape with a diameter that ranges from a few nanometers to tens of nanometers. Anyhow, the rolling direction is determined by the orientation of the graphene sheet, and it may be either a zigzag or armchair pattern. In general, this rolling direction determines the electronic properties of the nanotube such as its electrical conductivity and bandgap [239,240,241]. Single-wall CNTs (SWCNTs) have a single graphene layer that forms a cylinder, whereas multiwall CNTs (MWCNTs) have multiple graphene layers stacked on top of each other to form a cylinder within a cylinder. This gives MWCNTs a more complex structure and a larger diameter than SWCNTs. The structure of SWCNTs and MWCNTs can affect their properties. SWCNTs have a smaller diameter and a higher aspect ratio (length-to-diameter ratio) than MWCNTs, which contributes higher electrical and thermal conductivity [242,243]. On the other hand, MWCNTs have a larger diameter and a lower aspect ratio, which makes them great for mechanical reinforcement.

MXene and CNT nanocomposites have been shown as emerging candidates for energy harvesting. Specifically, they are used in the development of high-performance energy storage and conversion devices. The combination of MXene and CNT materials forms a unique material system with synergistic properties that are not found in either material alone. Evidently, the high electrical conductivity and high surface area of MXene can be combined with the high mechanical strength and stability of CNTs to create a nanocomposite with improved energy storage capabilities. The requirement of high energy density and high power density in energy harvesting devices can be fulfilled by the MXene–CNT composite materials [157,244,245].

Moreover, this involves the dispersion of CNTs and MXene in a solvent followed by the evaporation of the solvent to form the nanocomposite. The solvent can be chosen based on the solubility of the materials in the solution. In the in situ polymerization method, the CNTs and MXene are mixed together in a polymerization solution, and then the polymerization reaction is triggered to form the nanocomposite. The mechanical mixing method involves mechanically mixing the CNTs and MXene together followed by the compaction of the mixture to form the nanocomposite [117,246]. A solution containing both the CNTs and MXene was electrospun to form the nanocomposite in the co-electrospinning method. It is worth noting that the specific method used to make the nanocomposite will depend on the desired properties of the final material, as well as the compatibility of the CNTs and MXene with each other and with the method being used [247]. In addition, the helical host–guest structure of MXene/CNT hybrid fibers offers additional spaces for storing and transporting electrolyte ions and successfully inhibits the reaggregation of CNTs and MXene nanosheets.

Chuanwei Zhi et al. [248] developed a bio-inspired electronic skin design with directional moisture-wicking electronic skin with dual-mode sensing capability and biomechanical energy harvesting based on the construction of heterogeneous fibrous membranes and the controllable MXene/CNTs electrospraying layer. The morphology of electrosprayed MXene/CNTs ink (DMWES-3) on the C-PVDF substrate is shown in Figure 17a; the MXene nanosheets and CNTs were uniformly distributed on the hydrophobic layer while maintaining the porous structure of the hydrophobic layers. XRD responses of PVDF/MXene/CNT and the mechanical strength of the PVDF/MXene/CNT of the fabricated samples are illustrated in Figure 17b,c. The skin is comprised of a hydrophobic layer of CNT-modified PVDF nanofibers and a hydrophilic layer of PAN nanofibers, which is further enhanced with electrospraying MXene/CNTs-conductive ink to create heterogeneous wettability. The difference in hydrophobicity and hydrophilicity created a surface energy gradient that facilitated unidirectional moisture transfer resulting in a push–pull effect. This mechanism allows for the spontaneous absorption of sweat from the skin, ensuring the stability of bioelectrical signals. The electronic skin was developed and is shown in Figure 17d,e. The electronic skin with directional moisture-wicking properties demonstrated exceptional sensing abilities, including high sensitivity in a broad pressure range (up to 548.09 kPa^−1^), fast response and recovery times (28.4/39.1 ms), and remarkable static-sensing properties. Furthermore, the STENG employing the DMWES exhibited outstanding cycling stability and a high areal power density of 21.6 µW/m^2^ when utilized for energy harvesting from high-pressure sources (Figure 17f,g). By incorporating superior pressure sensing and triboelectric mechanism, the electronic skin exhibited exceptional capabilities for all-range healthcare sensing, including accurate pulse monitoring, gesture detection, and visual voice recognition. The study establishes a basis for the advancement of breathable self-powered electronic skins with high sensitivity, wide operating range, and multiple working modes. 

Xu Yang et al. [249] fabricated and developed the CNTs, 2D Nb_2_CTx MXene, and conductive PEDOT materials for obtaining MXene composites. The study investigated the impact of composition mass ratio on output performance, surface roughness, and stability of the resulting hybrid film. The researchers developed a flexible high-output TENG sensor by using a hybrid membrane as the electrode and triboelectric layer. The obtained sensor had a Voc of 184.1 V and an Isc of 4.42 μA. Additionally, the sensor displayed excellent sensitivity with a response time of 20 ms, a recovery time of 30 ms, and stable performance for over 400 cycles. Furthermore, the TENG sensor’s energy harvesting capability was confirmed by charging various capacitors. This discovery reveals a novel form of triboelectric material suitable for efficient energy harvesting as well as presents an uncomplicated approach for constructing stable electrode and triboelectric layers. This technology has great potential in the development of TENG-based wearable electronics. Hence, the integration of MXene and CNTs can be employed to design piezoelectric nanogenerators for energy harvesting applications. The excellent electrical conductivity of MXene and mechanical robustness of CNTs could enhance the overall piezoelectric characteristics of these devices. 

Md Salauddin et al. [250] chose to utilize the laser heating technique due to its cost-effectiveness, time-saving nature, and low power consumption for the production of the laser-carbonized (LC)-MXene/ZiF-67 nanocomposite. In contrast, the furnace method requires thousands of watts of power. Furthermore, thorough investigation and optimization of various factors have been conducted, including the MXene ratio on ZiF-67, laser power, laser speed, and number of passes. They evaluated these parameters by measuring the surface potential. Upon thorough analysis of the MXene ratio on ZiF-67, laser power, laser speed, and the number of passes, the optimized contact mode (CM)-TENG demonstrated significantly increased voltage and current density. Specifically, the voltage was found to be 13.4 times higher, while the current density showed a remarkable 14.5-fold enhancement compared to the single-layer configuration. Moreover, the contact and non-contact modes (CNM)-TENG were developed to possess exceptional qualities such as resistance to humidity, waterproofing, softness, skin-friendliness, and convenient compatibility with socks and shoes. Moreover, they ensure highly reliable and stable output performance, making them reliable choices for various applications. Upon attaching the device to the insole, the CM-TENG’s output performance was comprehensively studied during a range of human motions, including walking, running, jumping, and falling. The voltage signals from the CM-TENG were employed to analyze human gait and map foot movement patterns for individuals with varying characteristics. In addition, the CM-TENG exhibited the capability to power stopwatches, wristwatches, and digital hygrometers by harnessing the kinetic energy generated during human running motions. The results of this study present a novel and efficient approach for designing devices that showcase exceptional performance, making them highly suitable for applications in self-powered electronic devices and sensors.

Figure 18a provides a visual representation of the molecular structure of ZiF-67, MXene, and the fabrication process involved in synthesizing the LC-MXene/ZiF-67 nanocomposite. The Co^2+^ atoms within ZiF-67 form connections with imidazole anions through nitrogen atoms, resulting in a tetrahedral coordination structure. Furthermore, the field emission SEM (FESEM) image (Figure 18b) reveals an accordion-like multilayer morphology of MXene. Figure 18c–f presents the real-time signals captured by the NCM-TENG sensor integrated into the robot arm. These findings strongly suggest that the NCM-TENG holds great potential for the development of cost-effective, highly sensitive, and non-contact sensing systems. Particularly for vehicles and robots, NCM-TENG sensors play a crucial role in enabling obstacle avoidance capabilities.

Dongzhi Zhang et al. [251] introduced a PENG utilizing ZnO/MXene nanowire arrays as the driving force for a gas sensor. This gas sensor, based on an MXene/Co_3_O_4_ composite, demonstrated efficient detection of formaldehyde (HCHO) at room temperature (RT). The MXene/Co_3_O_4_ composite sensor exhibited clear selectivity and showcased excellent response characteristics specifically for HCHO detection. Furthermore, the influence of humidity conditions on HCHO detection demonstrates that the composite sensor holds great promise as a reliable candidate for real-time monitoring of HCHO at room temperature (RT). The improved response of the sensor to HCHO can be attributed to the synergistic interfacial interactions between MXene and Co_3_O_4_. Additionally, the PENG utilizing ZnO/MXene nanowire arrays has the potential to efficiently harness human motion energy, making it a promising candidate for the advancement of wearable devices for human use. The SEM and TEM images presented in Figure 19A provide additional evidence confirming the coexistence and binding of MXene and Co_3_O_4_. Moreover, the high-resolution transmission electron microscope (HRTEM) result displayed in Figure 19(Ac) reveals the distinct boundary between the two materials. Figure 19B illustrates the sensing characteristics of the sensor at various concentrations of HCHO under different angles. Figure 19(Bc) demonstrates the corresponding changes in resistance. Finally, this work exhibits the output voltage and response fitting of the MXene/Co_3_O_4_ self-powered sensor when subjected to air and HCHO at room temperature (RT), as shown in Figure 19D.

## 6. Three-Dimensionally Printed MXene Composites for Energy Harvesting

Three-dimensional printing is a promising technique for creating MXene composites for energy harvesting applications. Complex geometry can be created with precision, allowing for the design of custom-shaped structures that can optimize energy harvesting performance with 3D printing. Moreover, 3D printing allows for the creation of composite materials that incorporate MXene with other materials such as polymers or metals to create composites with enhanced properties. A 3D-printed MXene composite for energy harvesting can be a piezoelectric nanogenerator or triboelectric nanogenerator. These devices consist of a layer of MXene sandwiched between two layers of a piezoelectric polymer. When subjected to mechanical stress, the device generates an electrical current due to the piezoelectric properties of the polymer. The combination of MXene and 3D printing has the potential to create a new class of high-performance materials for energy harvesting applications.

Shipeng Zhang et al. [252] conducted a study on 3D-printed smart gloves for toroidal triboelectric nanogenerators through which the human–machine interaction was possible. A toroidal triboelectric sensor (STTS) has been developed which is self-powered and employs an MXene/Ecoflex nanocomposite with a fabric electrode to achieve high-output performance and flexibility. This sensor was designed to be used in scenarios involving human–machine interaction. To fabricate the wearable glove, a 3D printer was used to print the glove using TPU-95A, a flexible material. The TENG toroidal sensor package, connection wire package, and the fixation part for the back of the hand were printed separately and then assembled to complete the fabrication process of the wearable glove. The muscle releasing and tightening based the strain-inducing mechanism through the bending of fingers and flexible wearable gloves prepared by an MXene composite for generation of voltage are illustrated in Figure 20A. 

The effectiveness of the pyramidal structure in maintaining a gap between the positively charged finger skin and the negatively charged material layer was highly successful. This was achieved without the use of any spacer structure, allowing it to operate efficiently in contact separation mode. Due to its simple design, the single-electrode-based STTS can be easily worn on the finger for detecting finger movement without any need for fixation. This feature greatly improves its comfort and makes it easier to wear. The newly developed STTS exhibits remarkable performance with a Voc of 19.91 V, high sensitivity of 0.088 VkPa^−1^, and a wide pressure detection range of 0-120 kPa, which makes it suitable for the precise detection of finger movements in human–machine interaction scenarios (Figure 20B,C). This innovative technology has diverse applications in the next generation of smart and interactive products such as racing games, balance table control, appliance control, and robotic hand control. The newly developed self-powered intelligent glove showed great potential for the advancement of human–machine interaction technology. 

Qian Yi et al. [253] have successfully developed an innovative, flexible, and wearable 3D-printed sensing system based on MXene, which operates wirelessly and is self-powered. This system enables continuous and real-time monitoring of physiological signals, specifically focusing on the monitoring of RAP (respiratory acoustic pattern) waves. By seamlessly integrating M-TENG (mechanical triboelectric nanogenerator), M-PS (motion sensor), and multifunctional circuitry, the system efficiently generates power from mechanical motion and utilizes it for the monitoring of RAP waves in a continuous and real-time manner. MXene possesses an excellent triboelectric negative property, making it a highly suitable material for use in M-TENG. When coupled with SEBS (styrene-ethylene/butylene-styrene) in the M-TENG configuration, it exhibits an impressive output power of approximately 816.6 mW m^−2^. Through appropriate modifications, MXene demonstrates a threefold improvement in conductivity, along with a tunable viscoelastic property that is particularly advantageous for 3D printing applications. The M-PS (motion sensor) integrated within the MSP2 S3 system exhibits a remarkable sensitivity of approximately 6.03 kPa^−1^ and a rapid response time of about 80 ms. These features enable the detection of subtle changes in transient bio-signals with high precision. To the best of our understanding, this is the inaugural instance of a fully integrated wearable sensing system that utilizes MXene for triboelectric power generation. It operates self-powered and without the need for batteries, wirelessly enabling continuous and real-time monitoring of physiological signals powered by human motion. These groundbreaking devices hold immense promise for the future of wearable health monitoring, opening up exciting possibilities for advanced healthcare technologies.

Figure 21a,b illustrates the 3D-printed wearable sensing system based on MXene, demonstrating remarkable features. Its TENGs exhibit an impressive output power of approximately 816.6 mW m^−2^. The pressure sensors within the system showcase a sensitivity of 6.03 kPa^−1^, a low detection limit of about 9 Pa, and an impressive response time of around 80 ms. The voltage signal generated by the M-TENG when the elbow undergoes continuous bending is shown in Figure 21c. As presented in Figure 21d, showing the circuit diagram, the biomechanical energy that is derived from finger tapping is initially converted into electrical energy through the utilization of the M-TENG. Initially, the MSP2 S3 device was affixed to the arm of a volunteer and configured to enter the charging phase (Figure 21e,f). These responses were shown stage by stage as an NFC chip retrieves and retransmits the null value again, as shown in Figure 21g. Further development in the materials’ characteristics is still progressing. By tuning the molecular interactions between MXene and other constituents within the hydrogels, it was demonstrated that Ti_3_C_3_T_x_ MXene could act as a cross-linker to activate the fast gelation of hydrogels [254]. The mechanical properties, adhesion, and rapid shape transformation lead to the development of stretchable self-healing hydrogels [255]. The progress over these hydrogels based on MXene suggests that there could be great scope in the near future for MXene-based materials for energy harvesting applications through 3D printing technology [256,257]. Similarly, varieties of suitable novel materials are in the process of evaluation to be used in various nanogenerators. 

## 7. Conclusions and Outlook

A comprehensive review has been conducted on composite materials based on MXenes for TENGs and PENGs. The promising potential of nanogenerators is in their ability to harvest mechanical energy, function as multifunctional sensors, and more. The combination of MXenes with PVDF, PDMS, PTFE, PVA, Ag, Au, CNTs, and oxides, benefitting from their superior properties such as electronegativity, metallic conductivity, tunable surface chemistry, and mechanical flexibility, has been shown to positively impact the output performance of nanogenerators. The presence of MXenes in a material can result in a significant increase in electronegativity and conductivity, as well as facilitate fast electron transportation, provide excellent mechanical flexibility, regulate surface charge, and enhance cycling stability and durability. This review may provide guidance for the rational development of advanced nanogenerators that are capable of efficiently harvesting mechanical energy. 

This field is accompanied by some suggestions and perspectives put forth for consideration. In this field, many works have been completed for energy harvesting applications using polymers and polymeric composite materials. However, real-world problems still exist in analyzing natural systems to harvest green energy. Notably, MXene-based e-skin applications and wearable energy harvesting technologies are very promising areas of work that can be advanced to a higher level. The increasing demand for electronic devices and sensors necessitates the need for MXene-based nanogenerators with higher and more stable output performance to enable smart applications. Further in-depth studies can focus on the synthesis of MXenes, as the techniques used in the synthesis process can significantly impact the intrinsic properties and electrochemical output performance of MXenes. Factors such as flake sizes, defects, and surface chemistry are closely related to the synthesis techniques used. Additionally, several MXenes have only been predicted through theoretical calculations and have yet to be successfully synthesized. The development of additional MXenes and their derivatives can prove to be highly attractive as a source of materials for nanogenerators. 

## Figures and Tables

**Figure 1 micromachines-14-01273-f001:**
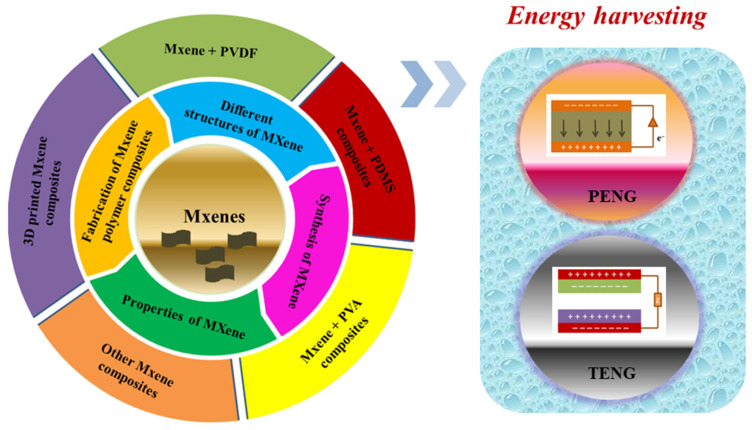
Overview of MXene-based materials for energy harvesting.

**Figure 2 micromachines-14-01273-f002:**
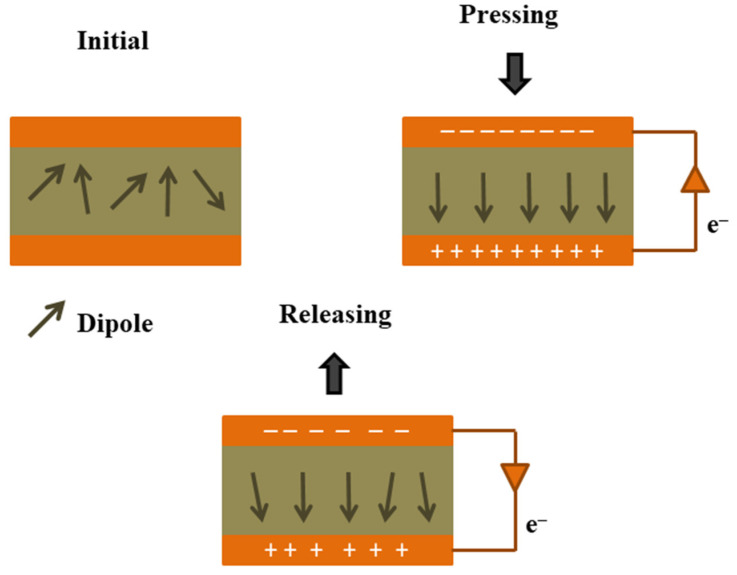
Schematic working mechanism of PENG.

**Figure 3 micromachines-14-01273-f003:**
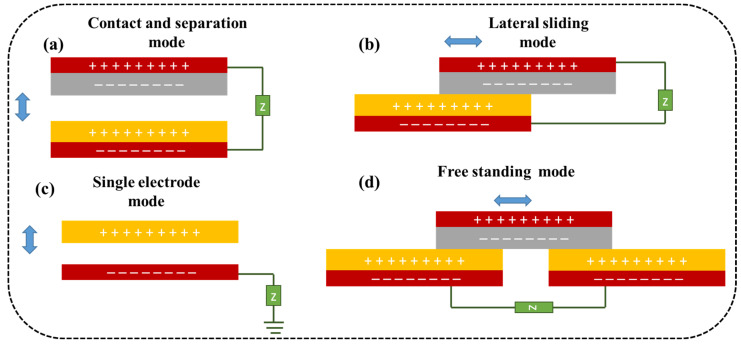
Schematic representation of modes of operation of TENG (**a**) contact and separation mode, (**b**) lateral sliding mode, (**c**) single electrode mode and (**d**) free standing mode.

**Figure 4 micromachines-14-01273-f004:**
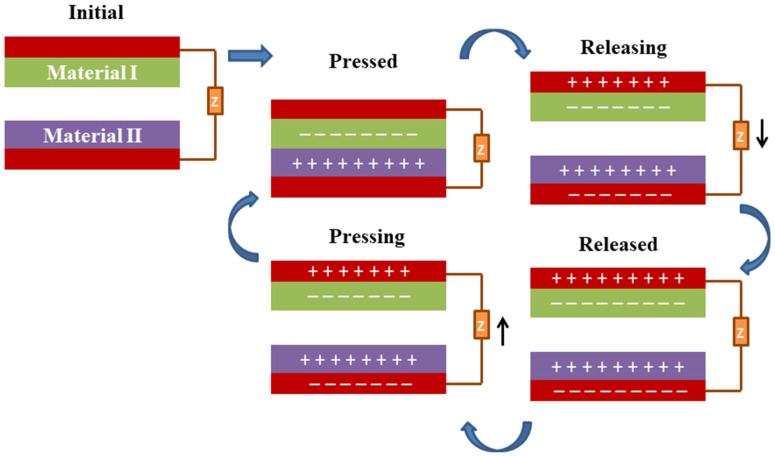
Schematic working mechanism of contact separation mode TENG.

**Figure 5 micromachines-14-01273-f005:**
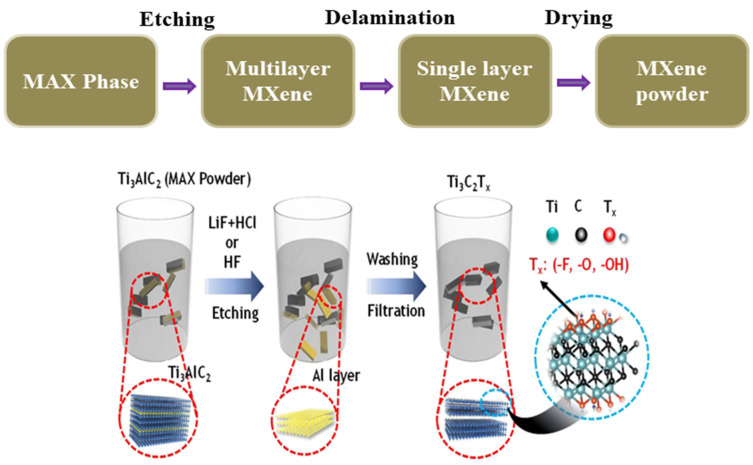
Synthesis procedure of MXenes through sonochemical method. Reprinted from [132], copyright (2020), permission conveyed through Copyright Clearance Center, Inc.

**Figure 6 micromachines-14-01273-f006:**
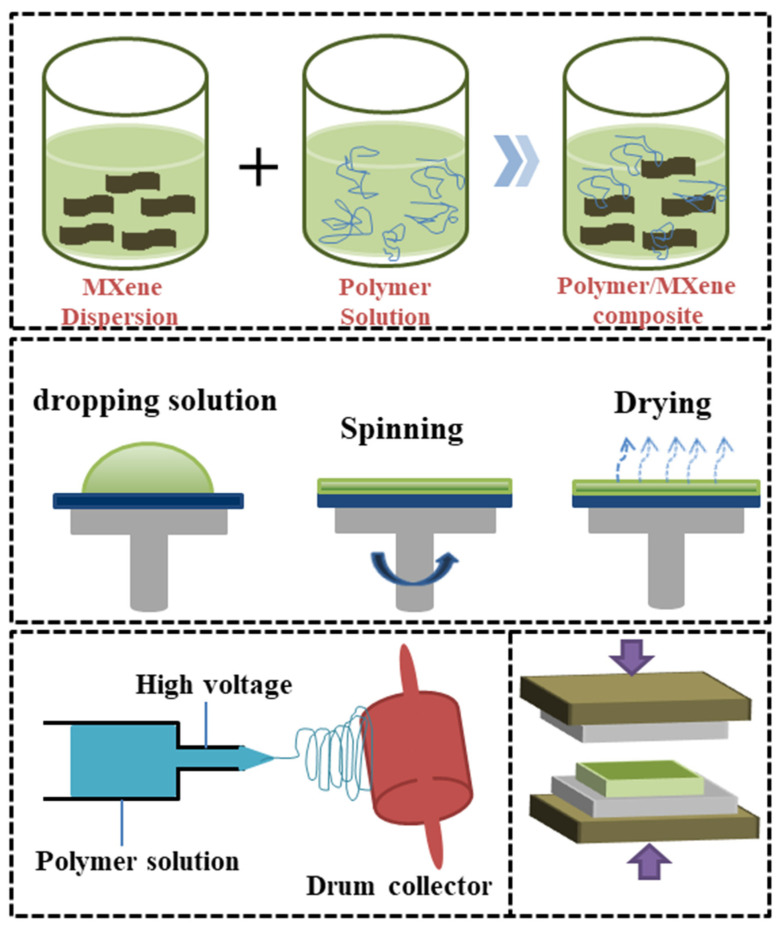
Fabrication process of MXene–polymer composites.

**Figure 7 micromachines-14-01273-f007:**
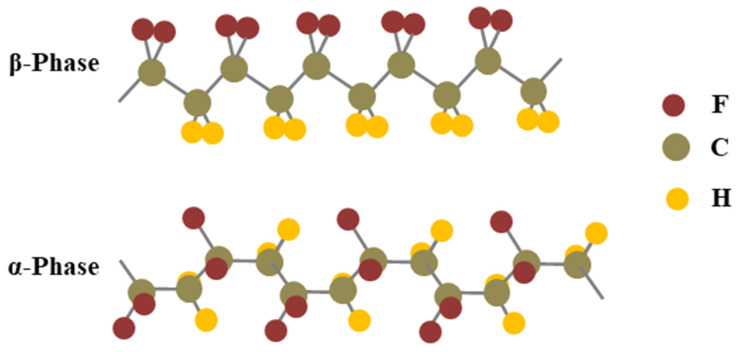
The molecular structure of PVDF β-phase and α-phase.

**Figure 8 micromachines-14-01273-f008:**
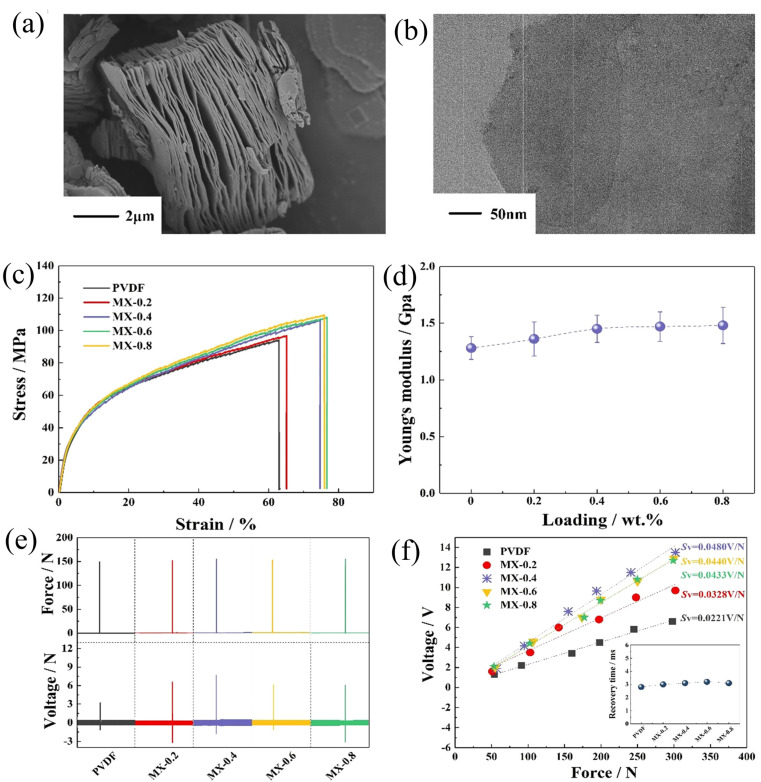
(**a**) SEM image of multilayer MXene, (**b**) TEM images of MXene nanosheets, (**c**) stress–strain curves, (**d**) Young’s modulus results of the composite samples, (**e**) output voltage, and (**f**) voltage vs. force for composites. Reprinted from [208], copyright (2021), with permission from Elsevier.

**Figure 9 micromachines-14-01273-f009:**
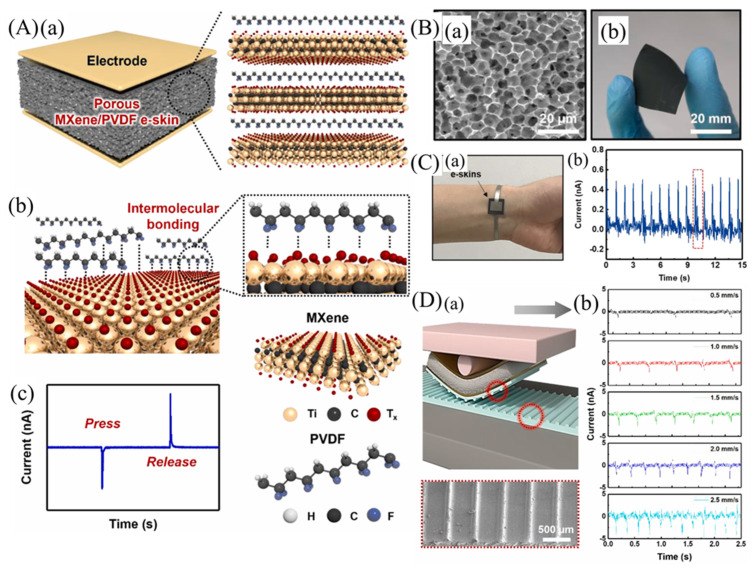
(**A**) (**a**) MXene multilayer composites model, (**b**) its molecular structures, and (**c**) piezoelectric output current. (**B**) (**a**) SEM images of porous structured MXene/PVDF e-skin and (**b**) Photograph of e-skin. (**C**) MXene-nanosheet-reinforced composites used as sensors and their output currents (**a**) Photograph of e-skins attached to the wrist for detecting radial artery pulse, (**b**) Real-time monitoring of radial artery pulse, and (**D**) E-skin-based energy harvesting and open-circuit voltage output (**a**) Schematic illustration of texture perception, wherein porous MXene/PVDF e-skins with PDMS fingerprint patterns (**b**) Time-dependent variation in piezoelectric currents when the patterned surface is scanned at different speeds (0.5–2.5 mm/s). Reprinted from [209], copyright (2021), with permission from Elsevier.

**Figure 10 micromachines-14-01273-f010:**
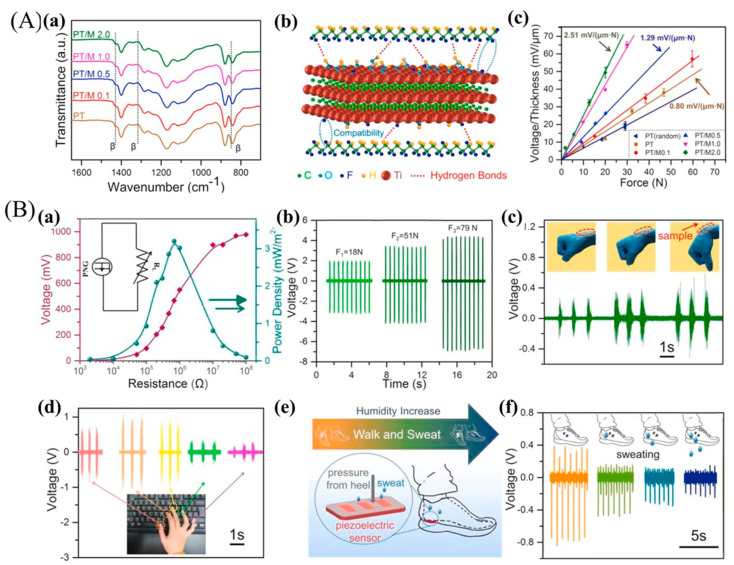
(**A**) (**a**) FTIR spectra of the PVDF-TrFE/MXene composites, (**b**) schematic interaction of MXene and polymer, and (**c**) linear fits of piezoelectric voltage output of PVDF-TrFE/MXene composite nanofibers with respect to force; (**B**) (**a**) power output of the composite fibers based on different resistance, (**b**) piezoelectric voltage output with respect to force applied on the PVDF-TrFE composites, (**c**) wearable energy harvesting which is tested by the wrist-induced strains, (**d**) voltage signals of the samples connected to the keys vary depending on the motions of different fingers when they touch them, (**e**) diagram illustrating the sensor that is placed underneath the heel to detect perspiration on the feet, and (**f**) voltage signals produced when pressure is applied to the heel during sweating of the foot. Reprinted from [211], copyright (2021), with permission from Elsevier.

**Figure 11 micromachines-14-01273-f011:**
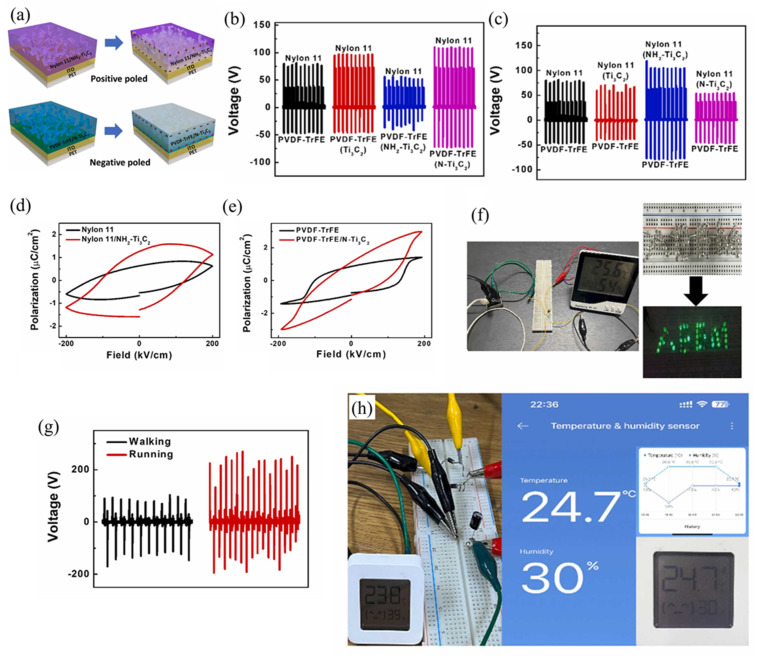
(**a**) Schematic representation showing the fabrication of positive and negative friction layers for TENGs. (**b**) Output voltage of PVDF-TrFE and PVDF-TrFE/Ti_3_C_2_, PVDF-TrFE/NH_2_-Ti_3_C_2_, and PVDF-TrFE/N-Ti_3_C_2_ composites as negative friction layers and Nylon 11 as the positive friction layer. (**c**) Output voltage of TENGs obtained using Nylon 11 and Nylon 11/Ti_3_C_2_, Nylon 11/NH_2_-Ti_3_C_2_, and Nylon 11/N-Ti_3_C_2_ composites as positive friction layers and PVDF-TrFE as the negative friction layer. (**d**,**e**) Polarization–electric field curves of Nylon 11/NH_2_-Ti_3_C_2_ and PVDF-TrFE/N-Ti_3_C_2_. (**f**) The photographs display a digital hygrometer and 50 green LEDs powered by TENGs. (**g**) TENGs fixed to the insole of a shoe can detect motion and harvest energy from walking or running motions. (**h**) TENG as humidity sensor. Reprinted from [216], copyright (2023), with permission from Elsevier.

**Figure 12 micromachines-14-01273-f012:**
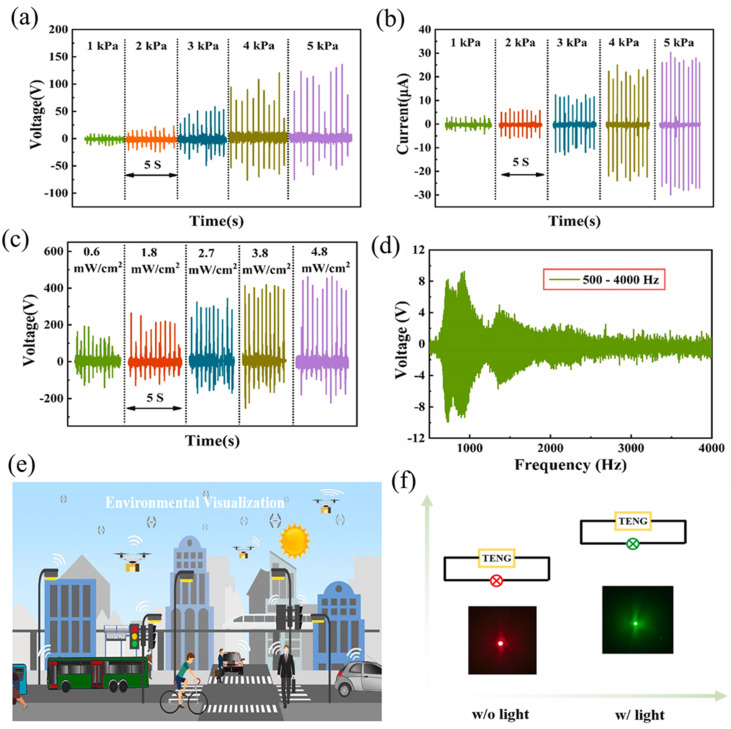
(**a**) Voc and (**b**) Isc of TENG with applied pressure, output voltage with (**c**) light intensity and (**d**) sound frequency, (**e**) a schematic diagram depicting the self-powered environmental visualized system, and (**f**) color of the LED changing to adapt to different environments. Reprinted from [227], copyright (2021), with permission from Elsevier.

**Figure 13 micromachines-14-01273-f013:**
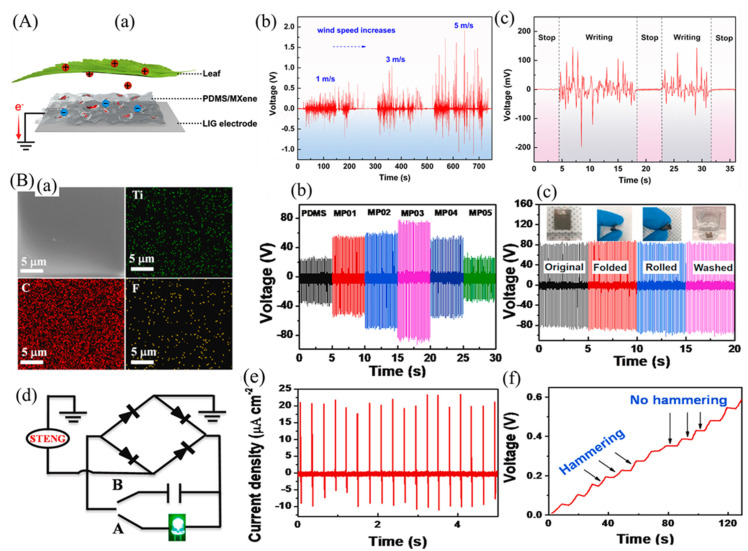
(**A**) (**a**) Working principle behind leaf swing energy collection is based on the coupling effect between contact electrification and electrostatic induction; (**b**) output voltage signal of TENG with the different wind speeds; (**c**) output voltage signal of writing motion, reprinted from [228], copyright (2019), with permission from Elsevier. (**B**) (**a**) EDX mapping of PDMS/MXene composite film, (**b**) output voltage of flexible STENG on MXene composition with the help of compression, (**c**) performance measurement after deformation or washing, (**d**) working electrical circuit of the flexible STENG-based self-charging system, (**e**) output current density for human-hammering-produced signals of flexible STENG, and (**f**) the charging of the capacitor during hammering action. Reprinted from [229], copyright (2020), with permission from Elsevier.

**Figure 14 micromachines-14-01273-f014:**
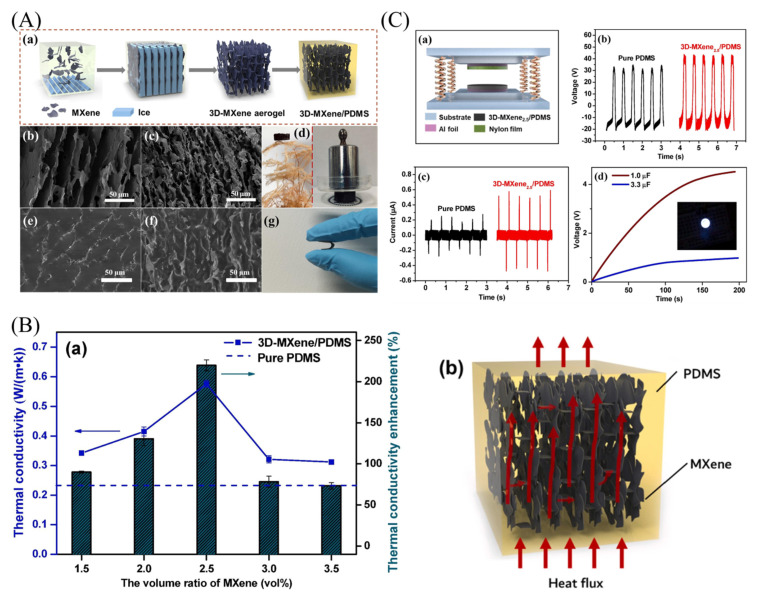
(**A**) (**a**) Fabrication procedure of 3D MXene/PDMS, (**b**,**c**) SEM images of prepared aerogel based 3D MXene, (**d**) digital image 3D MXene, (**e**,**f**) SEM images of 3D MXene2.0/PDMS and 3D-MXene2.5/PDMS nanocomposites, and (**g**) the flexibility of the PDMS/MXene composites. (**B**) (**a**) Through-plane thermal conductivities and enhancement of thermal conductivity at room temperature in 3D MXene/PDMS nanocomposites and (**b**) schematic diagram illustrating the improvement in through-plane heat transfer in 3D MXene/PDMS nanocomposites. (**C**) (**a**) schematic diagram of 3D-MXene/PDMS-based TENG, (**b**) output voltage, (**c**) output current of the 3D MXene/PDMS TENG, and (**d**) charging responses of the PDMS/MXene based on the commercial capacitors. Reprinted from [231], copyright (2020), with permission from Elsevier.

**Figure 15 micromachines-14-01273-f015:**
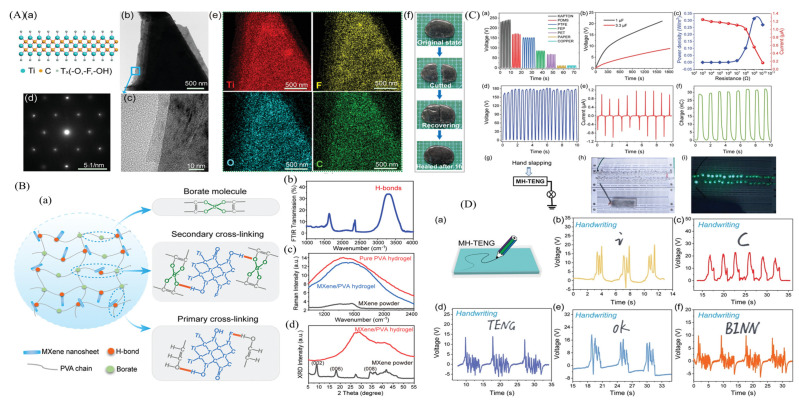
(**A**) (**a**) MXene/PVA hydrogel crystal structure, TEM image of the MXene nanosheets at (**b**) low magnification and (**c**) higher magnification, (**d**) SAED pattern, (**e**) EDX mapping results of the MXene nanosheets and (**f**) self-healing capability. (**B**) (**a**) The illustration of self-healing capability of MXene/PVA hydrogel and the primary and secondary cross-linking of MXene/PVA hydrogel, (**b**) XRD, and (**c**) Raman and (**d**) FTIR responses of the MXene/PVA hydrogel composite. (**C**) (**a**–**c**) Energy harvesting by MXene/Aerogel responses of open-circuit voltage, its charging behavior with 1 and 3.3 µF capacitors, and short-circuit current compared to others. (**d**–**f**) Voc, Isc, and transferred charge amount of the MH-TENG with hand slapping, (**g**) Schematic circuit diagram of lighted LEDs by the MH-TENG, (**h**) photograph of 40 LEDs connected with the MH-TENG without hand clapping, (**i**) The photograph of 40 LEDs lighted by hand tapping of the MH-TENG, and (**D**) Energy harvesting and handwritten detection by the different handwritten details for MH-TENG based on the different voltage responses (**a**) Illustration of handwriting on the surface of MH-TENG, and (**b**–**f**) Repeatable voltage signals for sensing different handwriting details. Reprinted from [234], copyright (2021), with permission from John Wiley & Sons.

**Figure 16 micromachines-14-01273-f016:**
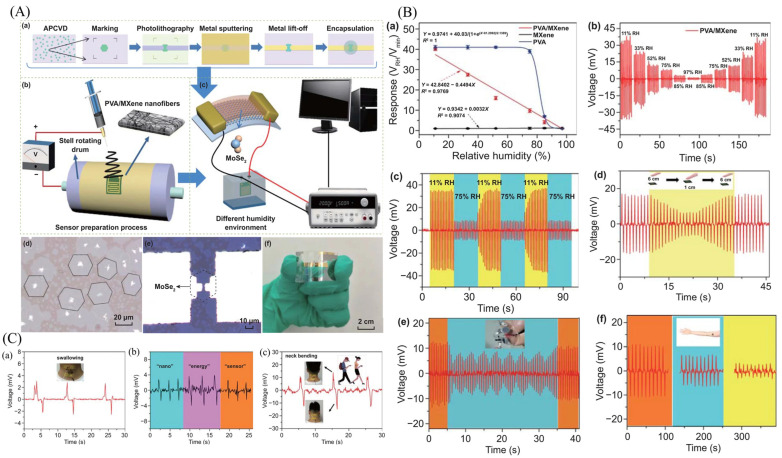
(**A**) (**a**) Schematic of fabrication of PENG with MoSe_2_, (**b**) Preparation of humidity sensor of PVA/MXene, (**c**) Schematic of the experimental platform for humidity sensing measurement, (**d**–**f**) illustration of prepared Monolayer MoSe_2_ by APCVD process and fabricated piezoelectric devices with two electrodes along with device of flexible PENG, (**B**) (**a**–**f**) output voltage responses of PVA and PVA/MXene based sensors responses for different humidity percentages, An illustration of the test results for detecting the humidity levels on the surface of the arm skin after different exercise durations, and (**C**) (**a**–**c**) piezoelectric responses of PVA/MXene exposed to swallowing condition and neck bending condition. Reprinted from [235] copyright (2021) permission conveyed through Copyright Clearance Center, Inc.

**Figure 17 micromachines-14-01273-f017:**
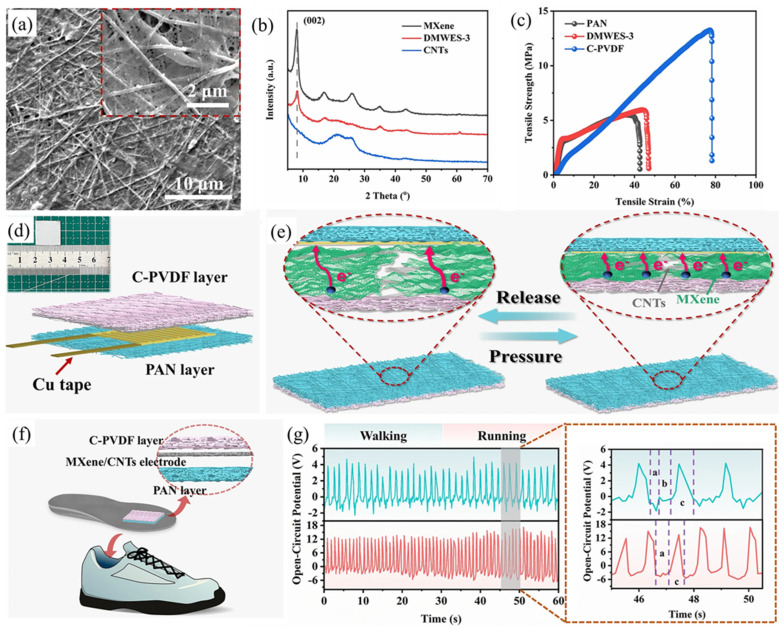
PVDF/CNT/MXene composite materials: (**a**) the illustration of SEM images of PVDF/MXene with CNT, (**b**) XRD responses of PVDF/MXene/CNT and (**c**) mechanical strength of the PVDF/MXene/CNT of the fabricated samples, (**d**) fabricated composite pressure-sensing elements, (**e**) pressure-sensing performance, and the sensing releasing and pressure applying mechanism, and (**f**) pictorial representation of pressure-sensing mechanism (**g**) signals from the pressure-sensing element while walking and running and its corresponding open-circuit voltages. Reprinted from [248], copyright (2023), permission conveyed through Copyright Clearance Center, Inc.

**Figure 18 micromachines-14-01273-f018:**
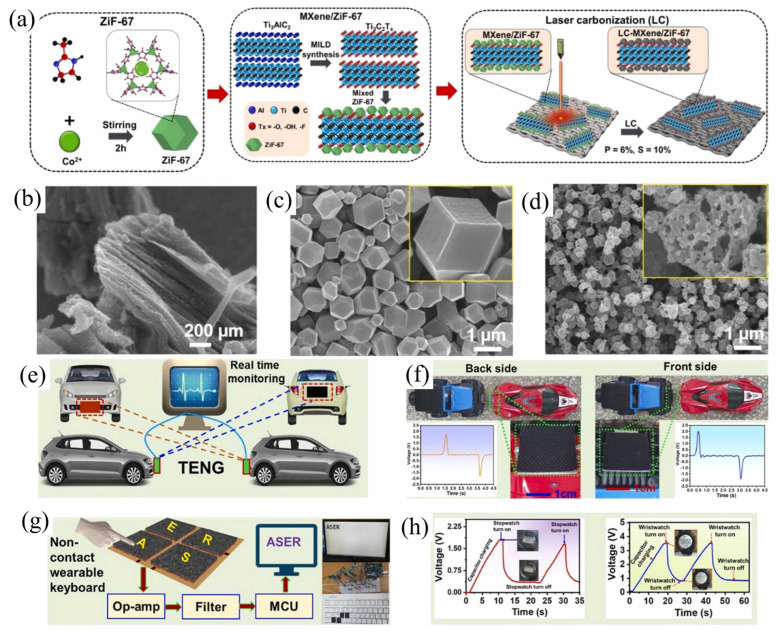
(**a**) Fabrication of ZiF-67, MXene, and LC-MXene/ZiF-67 nanocomposites, FESEM images of MXene with (**b**) ZiF-67, (**c**) LC-ZiF-67, and (**d**) LC-MXene/ZiF-67 (TENG application of prepared devices), (**e**) prevention of hitting obstacles in the robot, (**f**) collisions avoidance setup of NCM-TENG for the robot vehicle based on obtaining signals from front and back, (**g**) non-contact type of wearable keyboard based on using LC-MXene/ZiF-67, and (**h**) electronic devices powered by fabricated TENGs such as wristwatch and stopwatch with the voltage responses. Reprinted from [250], copyright (2022), with permission from Elsevier.

**Figure 19 micromachines-14-01273-f019:**
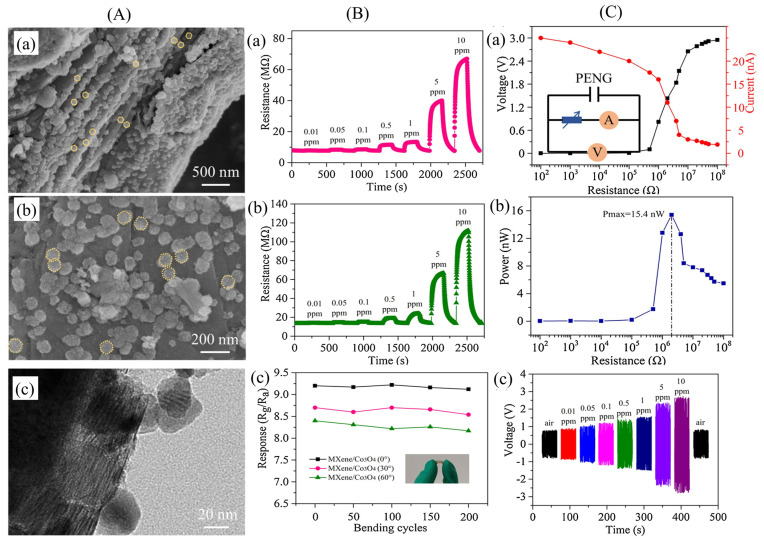
(**A**) (**a**) SEM images of MXene/Co_3_O_4_, (**b**) Co_3_O_4_, and (**c**) TEM image of MXene/ Co_3_O_4_. (**B**) The resistance changes in MXene/Co_3_O_4_ sensor with bending at (**a**) 30° and (**b**) 60°, (**c**) multiple times. (**C**) (**a**) Dependence of the output voltage/current (-circuit diagram), (**b**) relationship of the P vs. load resistance of PENG, and (**c**) sensor responses of MXene/Co_3_O_4_ driven by ZnO/MXene PENG while being exposed to the various concentrations and to the air. Reprinted with permission from [251], copyright (2021), Elsevier.

**Figure 20 micromachines-14-01273-f020:**
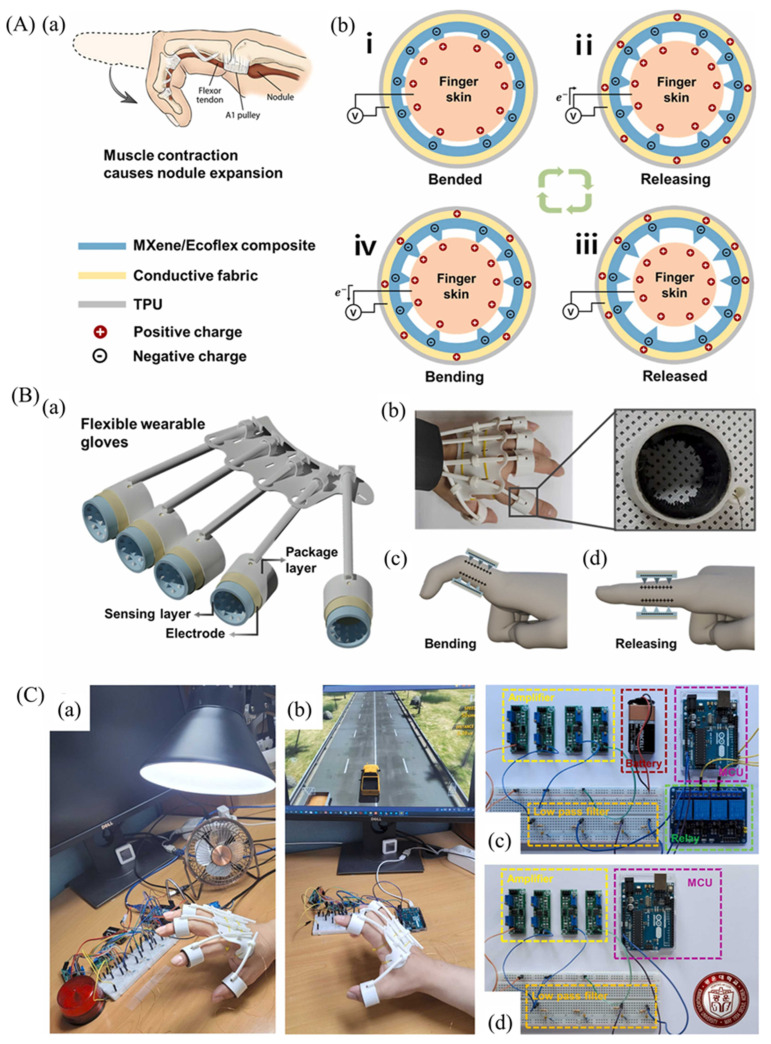
Wearable based on MXene composites: (**A**) (**a**) Bending the fingers caused muscle contraction and nodal expansion and (**b**) muscle releasing and tightening based strain-inducing mechanism through bending of fingers (**i**) In equilibrium, triboelectric charges with opposite polarity are induced, which does not result in an electron flow, (**ii**) free electrons to flow from the fabric electrodes to the ground (**iii**) electrostatic induction phase can produce an output voltage/current signal that eventually ends when the negative triboelectric charge on MXene/Ecoflex almost cancels out the induced positive charge on the fabric electrode, (**iv**) As the human skin approaches the MXene/Ecoflex, the induced positive charge on the fabric electrode decreases, and electrons flow from the ground to the fabric electrode until the human skin is in full contact again, (**B**) (**a**) flexible wearable gloves prepared by MXene composite for generation of voltage, (**b**–**d**) image of 3D-printed gloves and their working process during bending and releasing; (**C**) (**a**) schematic of the MXene-based self-powered physiological sensing system which is printed in 3D printing method for pressure sensor, (**b**) STTS is applied to the racing game, and (**c**–**d**) test circuit setup. Reprinted from [252], copyright (2023), with permission from Elsevier.

**Figure 21 micromachines-14-01273-f021:**
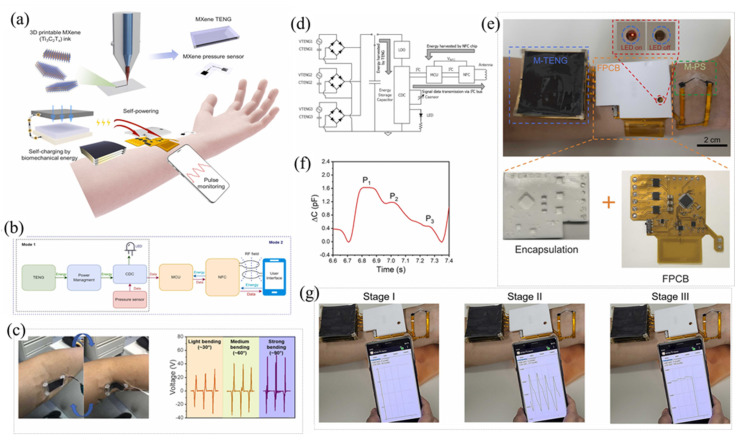
(**a**) Three-dimensionally printed MXene pressure sensor by home-modified MXene ink, power-management, and an energy-storing circuit and data collection, (**b**) experimentation with 3D-printed wearable pressure sensor and their output voltage of TENG as a response from bending of the elbow, (**c**,**d**) detection of the wrist pulse using application circuit through biomechanical energy as a representative physiological signal using only biomechanical energy, (**e**) optical image of the wearable device where the ON and OFF status is identified by the LED, (**f**) analyzing a particular pulse in which it shows the 3 distinctive peaks of a single wrist RAP pulse, and (**g**) the developed customized application for monitoring real-time pulse data with display. Reprinted from [253], copyright (2022), with permission from Elsevier.

## Data Availability

The data presented in this study are available from the corresponding author upon request.

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
