# Peer review of "MXene-Based Nanocomposites for Piezoelectric and Triboelectric Energy Harvesting Applications"

_micromachines, 2023, doi:10.3390/mi14061273_

Round 1
Reviewer 1 Report
In this manuscript, the author summarized the recent development of MXenes for the preparation of nanogenerators. The concept of energy harvesting was first introduced to stress the importance of renewable energy. Materials used for nanogenerators were also discussed. Finally, the synthesis of MXene, the preparation of MXene/polymer composite, and the energy harvesting performances of MXene composites were emphasized. I suggest the publication of this manuscript in Micromachines, a high-performance journal if the author can address the following concerns appropriately.
1. In the section “3. Materials used in nanogenerators”, what was the abbreviation of PVDF-HFP, PVDF-TrFE, PU, PVC, and PDMS? The author should define them first instead of using them directly. Besides, the author defined polydimethylsiloxane (PDMS) on Page 14 but used the abbreviation on Page 7. The author should be careful before writing down these terms. Thus, the author is suggested to check the full manuscript to avoid such errors. Besides, the author stressed the “Fabrication of MXene Polymer composite” in part 4.5 on Page 10, and “MXene /Polymeric Composites for Energy harvesting” was presented in part 5 on Page 11. The order organized here was weird and the author was suggested to double-check if there was any impropriety.
2. There are numerous kinds of polymers, however, only PVDF and PDMS were chosen as examples. Actually, PVDF and PDMS are general materials commonly seen in daily life, which may not highlight the novelty of this review. The author should give strong reasons for this.
3. MXene with high mechanical flexibility, metallic conductivity, and tunable surface property is a good candidate to be incorporated into the hydrophilic hydrogels to prepare stretchable yet conductive materials. Besides, a lower mechanical rigidness of hydrogels than elastomers induced easier performance tunability. The customizable rheological feature, cost-effectiveness, and well-defined structures also indicated the excellent combination of MXene and 3D printing for enhanced performances. To date, MXene hydrogels based on 3D printing (DOI: 10.1021/acsnano.0c07998; DOI: 10.1002/adfm.202107437; DOI: 10.3390/polym14101992; DOI: 10.3390/bios13050495) were reported. The author can compare the advantages and disadvantages of MXene hydrogels and MXene elastomers regarding the material synthesis and performance.
The English expression should be improved to raise the standard.
Author Response
Reviewer 1
In this manuscript, the author summarized the recent development of MXenes for the preparation of nanogenerators. The concept of energy harvesting was first introduced to stress the importance of renewable energy. Materials used for nanogenerators were also discussed. Finally, the synthesis of MXene, the preparation of MXene/polymer composite, and the energy harvesting performances of MXene composites were emphasized. I suggest the publication of this manuscript in Micromachines, a high-performance journal if the author can address the following concerns appropriately.
- In the section “3. Materials used in nanogenerators”, what was the abbreviation of PVDF-HFP, PVDF-TrFE, PU, PVC, and PDMS? The author should define them first instead of using them directly. Besides, the author defined polydimethylsiloxane (PDMS) on Page 14 but used the abbreviation on Page 7. The author should be careful before writing down these terms. Thus, the author is suggested to check the full manuscript to avoid such errors. Besides, the author stressed the “Fabrication of MXene Polymer composite” in part 4.5 on Page 10, and “MXene /Polymeric Composites for Energy harvesting” was presented in part 5 on Page 11. The order organized here was weird and the author was suggested to double-check if there was any impropriety.
Thanks for the comments. We have corrected the errors and the acronyms were introduced at very first time itself. Also, the section 4.5 is moved to 5.1 to keep the manuscript in a structure.
- There are numerous kinds of polymers, however, only PVDF and PDMS were chosen as examples. Actually, PVDF and PDMS are general materials commonly seen in daily life, which may not highlight the novelty of this review. The author should give strong reasons for this.
Thanks for this question. As mentioned by the reviewer, there are many polymers which can be utilized. As PVDF and PDMS are extensively studies and successfully utilized in demonstration of the possible application in energy harvesting, we have focused these two polymers only. We could have covered other polymers also in this review, but to reduce the length and make this this review article crisp, we have limited ourselves with these two polymers.
- MXene with high mechanical flexibility, metallic conductivity, and tunable surface property is a good candidate to be incorporated into the hydrophilic hydrogels to prepare stretchable yet conductive materials. Besides, a lower mechanical rigidness of hydrogels than elastomers induced easier performance tunability. The customizable rheological feature, cost-effectiveness, and well-defined structures also indicated the excellent combination of MXene and 3D printing for enhanced performances. To date, MXene hydrogels based on 3D printing (DOI: 10.1021/acsnano.0c07998; DOI: 10.1002/adfm.202107437; DOI: 10.3390/polym14101992; DOI: 10.3390/bios13050495) were reported. The author can compare the advantages and disadvantages of MXene hydrogels and MXene elastomers regarding the material synthesis and performance.
Thanks for the reviewers comments. We have included discussions on the suggested references in the revised manuscript.
Reviewer 2 Report
Review comments on micromachines-2441911-peer-review-v1
This review article deals with the MXene-based Nanocomposites for Piezoelectric and Triboelectric Energy Harvesting Applications. This systematic review covers the most recent developments of MXenes for nanogenerators, the importance of renewable energy, introduction to nanogenerator, major classifications and their working principles and so on. Moreover, it deals with MXene synthesis along with their properties, and MXene nanocomposites with polymeric materials are discussed in detail with the recent progress and challenges for their use in nanogenerator applications. Overall, this review article was well written and worthy for publication in micromachines. The following points may be considered before acceptance.
1. Most of cited references between 2020 and 2023, however, from the title “MXene-based Nanocomposites for Piezoelectric and Triboelectric Energy Harvesting Applications”, is the review focused on recent development on MXene-based Nanocomposites for Piezoelectric and Triboelectric Energy Harvesting Applications? or???
2. It is suggested to include some more method of Synthesis of MXenes under this section (4.3 Synthesis of MXenes)
3. Figure 5 caption may be revised (give the method of synthesis)
Minor editing of English language required
Author Response
Reviewer 2
This review article deals with the MXene-based Nanocomposites for Piezoelectric and Triboelectric Energy Harvesting Applications. This systematic review covers the most recent developments of MXenes for nanogenerators, the importance of renewable energy, introduction to nanogenerator, major classifications and their working principles and so on. Moreover, it deals with MXene synthesis along with their properties, and MXene nanocomposites with polymeric materials are discussed in detail with the recent progress and challenges for their use in nanogenerator applications. Overall, this review article was well written and worthy for publication in micromachines. The following points may be considered before acceptance.
- Most of cited references between 2020 and 2023, however, from the title “MXene-based Nanocomposites for Piezoelectric and Triboelectric Energy Harvesting Applications”, is the review focused on recent development on MXene-based Nanocomposites for Piezoelectric and Triboelectric Energy Harvesting Applications? or???
The evolution of MXene in different applications was started recently. There were articles on the development of energy harvesters in earlier years. However the reasonable development is showed very recently and the incorporation of MXene in optimized conditions took more time to bring out the better outputs. So the large numbers of references are from the recent times. However all the references are not belongs to the year between 2020-2023. The answer for the above query is, yes, we have discussed the recent progress and development of MXene based composites for Piezoelectric and Triboelectric Energy Harvesting Applications
- It is suggested to include some more method of Synthesis of MXenes under this section (4.3 Synthesis of MXenes)
Thanks for this suggestion. We have included a paragraph which is discussion on various synthesis methods of different MXenes in the revised manuscript
- Figure 5 caption may be revised (give the method of synthesis)
The figure caption is revised and the name of the synthesis process is included in the revised manuscript
Reviewer 3 Report
Review comments
MXenes is an important family of 2D materials, and their application for nanogenerators has produced many important results. The paper included the fundamentals of both PENG and TENG, discussed the advances of MXenes for nanogenerators, and provided perspectives on this area. I recommend the publication of this manuscript, although I have some suggestions for the author to consider.
1. What is different and unique about this manuscript with previous reviews relevant to MXenes-based materials for nanogenerators?
2. When the authors discuss the PENG, some important works were missing. For example, the first DC nanogenerator and the first nanogenerator with laterally packaged piezoelectric wires are important in the development of PENGs, and they are all missing from this manuscript.
3. In Section 4, more descriptions on the fundamental information of MXenes were provided. However, the authors did not well associate the characteristics of MXenes with the improvement of output, and a more in-depth analysis of the differences in the role of MXenes in different works is needed.
4. What is the effect of MXene on the nanogenerator?
5. There are errors in the legends, please check carefully. The Figure label is confusing, such as Figure 9 and Figure 10. Annotations for figures should remain in the same format. Do you want to use i, ii, iii, or a, b, c, A, B, C or a, b,c?
6. Some related works about MXene-based materials could be useful for discussion, such as 10.1016/j.nanoen.2022.107556, 10.1016/j.ensm.2022.03.045, 10.1016/j.mattod.2021.11.021, 10.1021/acsami.8b21893.
Moderate editing of English language required.
Author Response
Reviewer 3
MXenes is an important family of 2D materials, and their application for nanogenerators has produced many important results. The paper included the fundamentals of both PENG and TENG, discussed the advances of MXenes for nanogenerators, and provided perspectives on this area. I recommend the publication of this manuscript, although I have some suggestions for the author to consider.
- What is different and unique about this manuscript with previous reviews relevant to MXenes-based materials for nanogenerators?
Some of the review articles are there similar to this title but among them a few review articles are discussing the MXene for TENG or PENG, a few review articles discussing MXene composite fabrication with numerous materials for nanogenerators. In this review article we have specifically focused PVDF and PDMS based MXene nanocomposites and their potential utilization in TENG and PENG along with brief introduction of different nanogenerators, MXene synthesis and final output as a nanogenerators.
- When the authors discuss the PENG, some important works were missing. For example, the first DC nanogenerator and the first nanogenerator with laterally packaged piezoelectric wires are important in the development of PENGs, and they are all missing from this manuscript.
Thanks for the comment. As suggested by the reviewer, we have included some discussions on laterally packaged piezoelectric wires in the revised manuscript.
- In Section 4, more descriptions on the fundamental information of MXenes were provided. However, the authors did not well associate the characteristics of MXenes with the improvement of output, and a more in-depth analysis of the differences in the role of MXenes in different works is needed.
Thanks for the comment. We have included a few more paragraphs to provide in depth analysis of the differences in the role of MXene in the revised manuscript.
- What is the effect of MXene on the nanogenerator?
A high metallic conductivity and electronegativity of MXenes is advantageous for some of the polymers which can be used to make a flexible nanogenerators, for example PDMS. It is also proven that the introduction of MXene with proper content in PVDF-based polymer can achieve significantly enhanced dielectric constant. In some cases the MXene were utilized for the better adhesion. MXenes can significantly enhance the electronegativity and conductivity, facilitate fast electrons transportation, ensure outstanding mechanical flexibility, regulate surface charge, and improve cycling stability and durability. Overall the effect of MXene on the nanogenerator is depending on the other materials involved in the fabrication of nanogenerators. Some more discussion on the above is included in the revised manuscript.
- There are errors in the legends, please check carefully. The Figure label is confusing, such as Figure 9 and Figure 10. Annotations for figures should remain in the same format. Do you want to use i, ii, iii, or a, b, c, A, B, C or a, b,c?
Thanks for the comment. We have checked all the images and modified to follow the uniform number system.
- Some related works about MXene-based materials could be useful for discussion, such as 10.1016/j.nanoen.2022.107556, 10.1016/j.ensm.2022.03.045, 10.1016/j.mattod.2021.11.021, 10.1021/acsami.8b21893.
Thanks for the comment. We have included these suggested references in appropriate places with suitable discussion in the revised manuscript.
Reviewer 4 Report
It's unbelievable that an MXene review paper doesn't have references from Dr. Yury Gogotsi, Dr. Barsoum, and Dr. Naguib, who invented MXene and investigated it thoroughly from almost every aspect. This paper is just a stack of literature reviews and has improper self-citations.
Author Response
Reviewer 4
It's unbelievable that an MXene review paper doesn't have references from Dr. Yury Gogotsi, Dr. Barsoum, and Dr. Naguib, who invented MXene and investigated it thoroughly from almost every aspect. This paper is just a stack of literature reviews and has improper self-citations.
We already included some of the articles from Dr. Yuri Gogotsi, and Dr. Barsoum in the submitted manuscript. We think the reviewer did not check the manuscript properly. Reference number 99, 102,118, 119, 134,150, 151, and 154 are belongs to the above mentioned research group. We did not include any self-citation in this review article. But still the reviewer mentioned that we have improper self-citations. Anyway as suggested by the reviewer, we have included some of the articles from Dr. Naguib at suitable places with a brief discussion in the revised manuscript.
Round 2
Reviewer 1 Report
I suggest the acceptance of this manuscripts after the revision.
The English should be improved before online publication.
Author Response
Thanks for the positive comment.
Reviewer 3 Report
The manuscript has been significantly improved. It can be accepted after some minor revision.
1. The a,b, and c of Figure 10 should maintain the same font. The font style of a, b, c in Figure 10B is Arial, while a, b, c in Figure 10A is Times New Roman. Figure 20 has the same issues. Please revise it.
2. In Figure 15, the (B) should be corrected as (D). The order of a,b,c,d,e, and f in Figure 21 is a bit confusing. Please double check your manuscript.
3. The copyright permission in the Figure captions should be added.
Author Response
Thanks for the comment. We have revised the manuscript as per your suggestions.
- The a,b, and c of Figure 10 should maintain the same font. The font style of a, b, c in Figure 10B is Arial, while a, b, c in Figure 10A is Times New Roman. Figure 20 has the same issues. Please revise it.
The font style has been changed in the revised manuscript.
- In Figure 15, the (B) should be corrected as (D). The order of a,b,c,d,e, and f in Figure 21 is a bit confusing. Please double check your manuscript.
Thanks for the comment. We have modified the figures in the revised manuscript.
- The copyright permission in the Figure captions should be added.
We have added the copyright permission information in the figure caption in the revised manuscript.
Reviewer 4 Report
good
good
Author Response
Thanks for the positive comment. we have improved and submitting the revised article.